# RELAYFORMER: A UNIFIED LOCAL-GLOBAL ATTENTION FRAMEWORK FOR SCALABLE IMAGE AND VIDEO MANIPULATION LOCALIZATION

**Wen Huang**[1], **Jiarui Yang**[2], **Tao Dai**[3]*, **Jiawei Li**[4], **Shaoxiong Zhan**[1], **Bin Wang**[1], **Shu-Tao Xia**[1]

[1]Tsinghua Shenzhen International Graduate School, Tsinghua University
[2]College of Artificial Intelligence, Nankai University
[3]College of Computer Science and Software Engineering, Shenzhen University
[4]Huawei Technologies Co., Ltd
`huang-w24@mails.tsinghua.edu.cn, daitao.edu@gmail.com,`
`xiast@sz.tsinghua.edu.cn`

## ABSTRACT

Visual manipulation localization (VML) aims to identify tampered regions in images and videos, a task that has become increasingly challenging with the rise of advanced editing tools. Existing methods face two central issues. The first is resolution diversity. Resizing or padding can distort subtle forensic cues and introduce unnecessary computational cost. The second is the difficulty of extending spatial models for images to spatio-temporal inputs in videos, which often results in maintaining separate architectures for the two data types. To address these challenges, we propose RelayFormer, a unified framework that adapts to varying resolutions and naturally handles both static and temporal visual data. Relay-Former partitions inputs into fixed-size sub-images and introduces Global Local Relay (GLR) tokens that propagate structured context through a relay-based attention mechanism. This design enables efficient exchange of global cues, such as semantic or temporal consistency, while preserving fine-grained manipulation artifacts. Unlike prior approaches that depend on uniform resizing or sparse attention, RelayFormer scales to variable resolutions and video sequences with minimal overhead. Experiments across diverse benchmarks demonstrate superior performance and strong efficiency, combining resolution adaptivity without interpolation or excessive padding, unified processing for images and videos, and a favorable balance between accuracy and computational cost. Code is available at https://github.com/WenOOI/RelayFormer.

## 1 INTRODUCTION

Visual manipulation localization (VML), covering both static images and temporally extended videos, is a fundamental task in digital forensics. Its goal is to precisely identify tampered regions within visual content. With the rapid proliferation of advanced editing tools, detecting and localizing such manipulations has become increasingly challenging (see Fig. 1(a)).

While prior research has predominantly focused on improving detection performance (Zhu et al., 2025; Lou et al., 2024), robustness (Guillaro et al., 2023; Qu et al., 2023), generalization (Zhou et al., 2023; Lou et al., 2025), and interpretability (Qu et al., 2024a) either for static images or for videos, existing methods still face two key limitations that hinder their applicability in real-world scenarios. **First**, resolution diversity poses a significant challenge. In-the-wild content ranges from low resolution (e.g., 256×256) to 4K. Unlike in standard vision tasks, interpolation can destroy the subtle low-level traces crucial for forensic analysis (Guillaro et al., 2023; Ma et al., 2023). Prior works rely on fixed-resolution training, forcing a trade-off: down-sampling inputs to a uniform size

---

*Corresponding authors: Tao Dai.
This work is supported in part by the National Natural Science Foundation of China, under Grant (62302309, 62571298), Tsinghua SIGS KA Cooperation Fund.

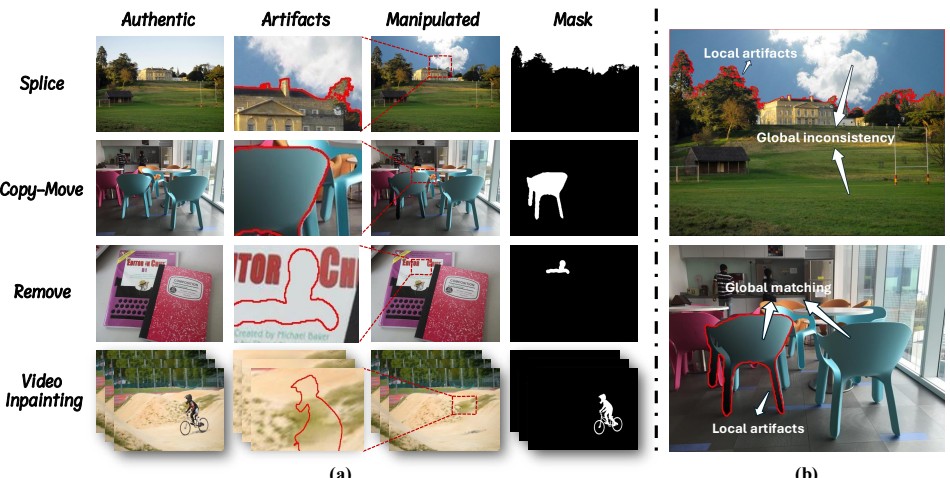

Figure 1: Illustration of several common types of visual manipulation, including splicing, copy-move, and inpainting. (a) Examples of manipulated regions and their corresponding boundaries generated by these methods. (b) A schematic illustration highlighting the need for both local and global information to accurately localize manipulated regions.

(e.g., 512×512) (Guillaro et al., 2023; Su et al., 2025), which risks losing manipulation artifacts, or padding smaller inputs to a large canvas (e.g., 1024×1024) (Ma et al., 2023), which incurs substantial computational redundancy. Furthermore, uniform resizing disproportionately distorts content with non-standard aspect ratios (e.g., 9:19.5 in modern smartphones), further compromising forensic reliability.

**Second**, the extension from static to temporal inputs introduces a modeling gap. Images and videos belong to the same visual modality, but videos add a temporal dimension that requires reasoning over frame-to-frame consistency. Existing algorithms are usually designed either for purely spatial inputs or for spatio-temporal inputs. Image-oriented models cannot leverage temporal cues, whereas video-oriented models often struggle to generalize to single images. This limitation forces practitioners to maintain two separate models, increasing both computational cost and system complexity.

Manipulation localization in images and videos demands a delicate balance between fine-grained sensitivity and global semantic reasoning. Manipulated regions are typically small and visually subtle, yet their reliable detection often hinges on scene-level consistency cues such as illumination patterns, object semantics, or temporal coherence across frames. Although dense global attention can, in principle, capture such dependencies, it is computationally prohibitive for high-resolution content. As illustrated in Fig. 1(b), the global cues essential for manipulation detection are relatively coarse, reflecting scene-level regularities rather than exhaustive pixel-level correspondence. For example, in the splicing case (top right), inconsistencies often manifest as illumination mismatches across the scene; in the copy–move case (bottom right), beyond local artifacts, detection relies on structural redundancy between the duplicated region and its source. These characteristics suggest that sparse yet effective global information propagation is both sufficient and desirable.

Building on this insight, we propose RelayFormer, a unified, efficient, and flexible architecture for VML. The key idea is to leverage structured global–local interactions without incurring the prohibitive cost of dense attention. RelayFormer dynamically partitions inputs into fixed-size sub-images according to resolution and introduces Global–Local Relay (GLR) tokens that mediate information exchange through a relay-based attention mechanism. Acting as information bottlenecks, these tokens iteratively absorb scene-level consistency cues, transmit compressed semantics across the entire sample, and reinject enriched context back into their respective regions. Unlike prior approaches (Yang et al., 2021; Su et al., 2025) that reduce computation primarily via sparse attention, the proposed architecture dynamically allocates computation according to input resolution while enabling task-oriented global information propagation. This design ensures scalability to variable resolutions and a natural extension from static images to temporal sequences.

To comprehensively validate the effectiveness of our framework, we conduct extensive experiments on a wide range of widely used benchmarks covering both static and temporal visual data. We further provide detailed quantitative and qualitative analyses demonstrating that our method consistently achieves superior performance while maintaining efficiency across diverse settings.

Our main contributions are as follows:

- **Resolution adaptivity.** We dynamically handle variable input resolutions without interpolation while substantially reducing redundant padding, preserving subtle forensic traces.
- **Unified image–video modeling.** We use a single architecture that naturally supports both spatial (image) and spatio-temporal (video) manipulation localization.
- **Efficient global–local reasoning.** We introduce GLR tokens to propagate structured global context efficiently without relying on dense full-resolution attention.

## 2 RELATED WORK

### 2.1 IMAGE MANIPULATION LOCALIZATION

Image-level approaches primarily differ in the forensic cues they exploit. Artifact-based methods (Wu et al., 2022; Wu & Zhou, 2021) detect low-level traces such as noise inconsistencies or compression residuals. Although effective in controlled settings, these cues are easily disrupted by common post-processing operations such as resizing or recompression, which leads to unstable performance in real-world scenarios. Another line of work leverages contrastive learning or structural priors. FOCAL (Wu et al., 2023) and NCL (Zhou et al., 2023) leverages contrastive learning to improve generalization ability, while our work is complementary as it focuses on efficiently handling dynamic resolutions and mixed image–video inputs. IML-ViT (Ma et al., 2023) demonstrates that high-resolution vision Transformers benefit from edge supervision, but full-resolution attention incurs high memory cost and limits scalability to diverse or larger inputs. Multi-scale architectures, including Mesorch (Zhu et al., 2025), improve robustness by enlarging receptive fields or combining convolutional and Transformer features. However, their reliance on high-resolution processing introduces substantial computational overhead. Fusion-based models such as TruFor (Guillaro et al., 2023) and CAT-Net (Kwon et al., 2022) combine RGB information with noise fingerprints or DCT-domain cues, improving the reliability of forensic evidence. Their performance, however, is constrained by assumptions tied to specific compression characteristics, which reduces their applicability to images from heterogeneous acquisition pipelines.

### 2.2 VIDEO MANIPULATION LOCALIZATION

Video manipulation localization extends spatial analysis to incorporate temporal information. VideoFACT (Nguyen et al., 2024) enriches spatial representations with contextual embeddings through self-attention, but its quadratic complexity restricts the feasible temporal length. ViLocal (Lou et al., 2025) utilizes contrastive learning to detect local spatiotemporal inconsistencies. In contrast, UVL2 (Pei, 2023) integrates cues such as spatial edges and global pixel distributions within a hybrid design to achieve robust localization. These methods achieve strong generalization yet remain computationally demanding due to dense temporal sampling or high-resolution spatial processing. VIDNet (Zhou et al., 2021) integrates RGB features with error level analysis (ELA) cues through a ConvLSTM decoder, although the reliance on ELA makes the method sensitive to modern re-encoding pipelines and common real-world perturbations.

## 3 METHOD

We present **RelayFormer**, a unified and modular framework for Visual Manipulation Localization (VML) that scales to variable image resolutions and temporal lengths. The framework is composed of three main components: *Input Unification*, *Global-Local Relay Attention*, and a *Query-based Mask Decoder*. These components together enable efficient spatial-temporal reasoning by balancing global consistency with local expressivity, while ensuring computational scalability.

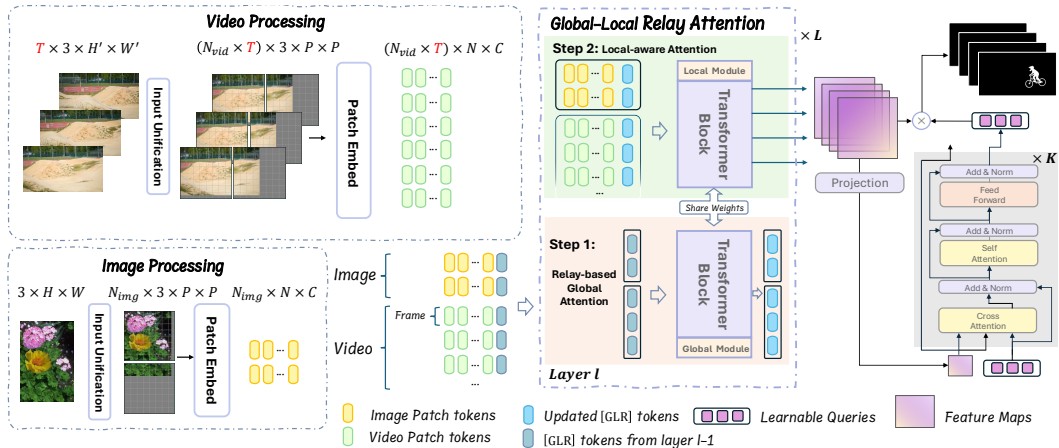

Figure 2: Overview of our proposed framework, which consists of three main components. First, the input image or video is partitioned into unified local sub-images without interpolation, preserving fine-grained spatial details. Second, we propose the GLRA module to achieve efficient global information propagation. Finally, a carefully designed lightweight mask decoder efficiently produces the prediction masks. For clarity, the positional encoding components are omitted from the figure.

## 3.1 INPUT UNIFICATION

To unify image and video inputs into a common representation suitable for parallel computation, we decompose all inputs into slightly overlapping local sub-images, which serve as the atomic processing elements in our framework.

**Image inputs.** Given an image $x \in \mathbb{R}^{C \times H_{\text{img}} \times W_{\text{img}}}$, we partition it into slightly overlapping sub-images of spatial size $H_p \times W_p$. Let the sliding strides along height and width be $S_h$ and $S_w$. Padding is applied if the remaining region is smaller than a full sub-image. The number of sub-images along each spatial dimension is

$$N_h = \left\lceil \frac{H_{\text{img}} - H_p}{S_h} \right\rceil + 1, \quad N_w = \left\lceil \frac{W_{\text{img}} - W_p}{S_w} \right\rceil + 1,$$

so the total number of sub-images for the image is $N_{\text{img}} = N_h \times N_w$. The resulting tensor has shape $(N_{\text{img}}, C, H_p, W_p)$.

**Video inputs.** For a video $x \in \mathbb{R}^{T \times C \times H_{\text{vid}} \times W_{\text{vid}}}$, we first merge the batch and temporal dimensions, treating the video as $(T, C, H_{\text{vid}}, W_{\text{vid}})$. Each frame is partitioned in the same way as images, producing $N_{\text{vid}} = N_h \times N_w$ sub-images per frame. The resulting tensor has shape $(T \cdot N_{\text{vid}}, C, H_p, W_p)$.

**Unified representation.** Finally, all sub-images from images and videos are concatenated into a batch of shape

$$(B_{\text{total}}, C, H_p, W_p),$$

where $B_{\text{total}} = \sum_{\text{images}} N_{\text{img}} + \sum_{\text{videos}} T \cdot N_{\text{vid}}$. Each sub-image is treated as an independent sample in the subsequent local modeling stage, enabling large-batch parallel computation without explicitly distinguishing between image and video inputs. We provide pseudocode in the Appendix A.2.1.

## 3.2 GLOBAL-LOCAL RELAY ATTENTION (GLRA)

To balance efficiency and expressiveness, we propose **Global-Local Relay Attention (GLRA)**, which enables efficient propagation of global context through a small set of learnable tokens, while retaining fine-grained local modeling. Fig. 3 shows the detailed structure of GLRA.

**Local-aware Attention.** For each sub-image $U_i$, we apply a ViT patch embedding to obtain patch tokens $X_i \in \mathbb{R}^{P \times d}$, where $P$ is the number of tokens and $d$ is the feature dimension. We append a small set of learnable Global-Local Relay [GLR] tokens $T_i \in \mathbb{R}^{m \times d}$ to each sub-image:

$$[T_i^{(l)}, X_i^{(l)}] = \text{SelfAttn}_{\text{local}}([T_i^{(l-1)}; X_i^{(l-1)}]), \tag{1}$$

where $l = 1, \ldots, L$ represents the layer. In this stage, the [GLR] tokens both relay global information obtained from previous layers and absorb localized details from $X_i$.

**Relay-based Global Attention.** To enable global information exchange, we aggregate [GLR] tokens from all sub-images:

$$T_{\text{flat}} = \text{Concat}_{j=1}^{N_i} T_j \in \mathbb{R}^{(N_i \cdot m) \times d}, \tag{2}$$

where $N_i$ denotes the number of sub-images in the sample. Each [GLR] token is encoded with temporal index, spatial location, and token identity using 4D Rotary Positional Embeddings (RoPE) (Su et al., 2024; Wang et al., 2024). The global attention step is then:

$$T_{\text{updated}} = \text{SelfAttn}_{\text{global}}(\text{RoPE}_{4D}(T_{\text{flat}})). \tag{3}$$

After global attention, the updated [GLR] tokens are injected back into their corresponding sub-images, enabling iterative information relay: 1) in the local attention stage, [GLR] tokens transmit global context into local sub-images while gathering new local evidence; 2) in the global attention stage, they exchange these enriched representations with [GLR] tokens of other sub-images.

**Parameter-efficient strategy.** Using shared parameters for local sub-module and global sub-module would reduce performance because they have conflicting goal. Shared weights lead to poor performance in both. While conceptually separating local and global attention into two distinct Transformer layers is straightforward, this naive approach doubles the parameter count overhead for each such block. Our core motivation stems from the hypothesis that the computational processes for local and global attention, while functionally distinct, share a substantial underlying structure. To capitalize on this insight, we propose a parameter-efficient strategy. We maintain a single, shared Transformer backbone layer for both the local and global attention computations. To induce the necessary functional specialization, we introduce two distinct adaptation modules (e.g., LoRA (Hu et al., 2022) or Adapters (Poth et al., 2023)), one for each attention mechanism. Specifically, the shared backbone layer learns the common, foundational features of the attention mechanism. The adaptation module for local attention learns the specific residual transformation required to specialize the shared function for processing fine-grained patterns, while the module for global attention learns the residual required for long-range, contextual reasoning. This approach allows us to achieve the expressive power and performance nearly identical to a two-layer model, but with only a marginal increase in parameters over a single-layer baseline, thereby achieving a superior trade-off between performance and efficiency. We provide more implementation details of this in the Appendix A.2.2.

**4D RoPE Formulation and Extrapolation.** We decompose the hidden dimension of each token into five groups: temporal ($T$), token index ($id$), vertical ($H$), horizontal ($W$), and the remaining. Specifically, for a token vector:

$$x = [x_T, x_{id}, x_H, x_W, x_{rem}],$$

where $x_T \in \mathbb{R}^{d_T}, x_{id} \in \mathbb{R}^{d_{id}}, x_H \in \mathbb{R}^{d_S}, x_W \in \mathbb{R}^{d_S}, x_{rem} \in \mathbb{R}^{d_{rem}}$, with $d_T + d_{id} + 2d_S + d_{rem} = d$.

For each group, a standard 1D RoPE (Su et al., 2024) is applied independently with the corresponding positional index (temporal id, token id, height id, width id). Formally, for a sub-vector $x_g \in \mathbb{R}^{d_g}$ and index $p_g$, we apply:

$$\text{RoPE}(x_g^{(2i)}, x_g^{(2i+1)}) = \begin{bmatrix} x_g^{(2i)} \cos(p_g \theta_i) - x_g^{(2i+1)} \sin(p_g \theta_i) \\ x_g^{(2i)} \sin(p_g \theta_i) + x_g^{(2i+1)} \cos(p_g \theta_i) \end{bmatrix},$$

where $\theta_i = 10000^{-2i/d_g}$.

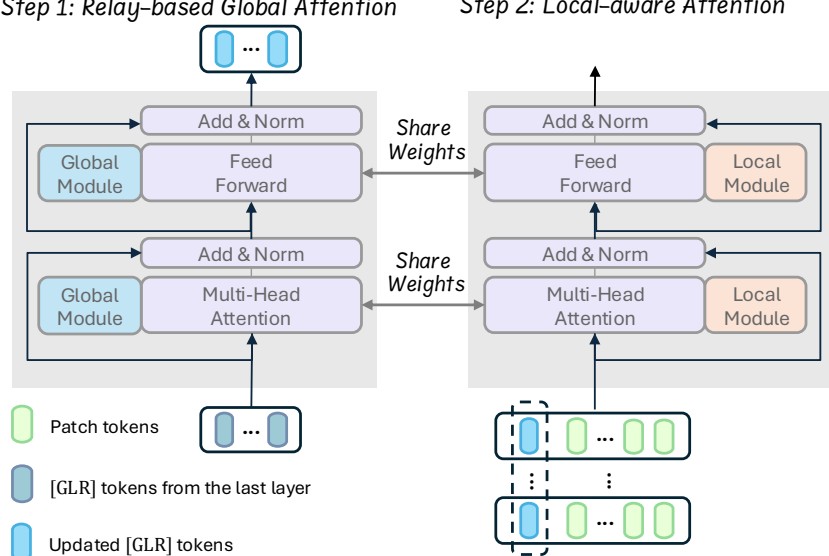

Figure 3: Detailed architecture of the proposed Global-Local Relay Attention (GLRA) module.

The final rotated embedding is:

$$\mathrm{RoPE}_{4D}(x) = [\mathrm{RoPE}(x_T), \mathrm{RoPE}(x_{id}), \mathrm{RoPE}(x_H), \mathrm{RoPE}(x_W), x_{rem}].$$

This formulation applies independent rotary encodings across temporal, token index, and spatial dimensions, equipping our model with strong extrapolation capabilities to variable resolutions.

### 3.3 QUERY-BASED MASK DECODER

To avoid decoding becoming a computational bottleneck, we design a lightweight query-based Transformer decoder, inspired by Mask2Former (Cheng et al., 2022). Given the reassembled feature map $F \in \mathbb{R}^{H_f \times W_f \times d}$, we first project it into a lower-dimensional space $\tilde{F} \in \mathbb{R}^{H_f \times W_f \times d_{low}}$. A small set of learnable queries $Q \in \mathbb{R}^{M_f \times d}$ then interacts with the projected feature map.

The decoder is composed of $K$ stacked layers. At the $k$-th layer ($k = 1, \ldots, K$), query features are updated via a cross-attention followed by a self-attention operation:

$$Q^{(k)'} = \mathrm{CrossAttn}(Q^{(k-1)}, \tilde{F}), \tag{4}$$

$$Q^{(k)} = \mathrm{SelfAttn}(\mathrm{RoPE}(Q^{(k)'})). \tag{5}$$

Finally, a gating MLP assigns weights to each query, modulating its contribution to the predicted manipulation masks.

### 3.4 LOSS FUNCTION

Following previous methods (Ma et al., 2023), we adopt a combination of binary cross-entropy (BCE) loss and edge loss. The overall loss is defined as:

$$\mathcal{L} = \mathcal{L}_{\mathrm{BCE}}(P, M) + \lambda \cdot \mathcal{L}_{\mathrm{Edge}}(P \odot M_e, M \odot M_e) \tag{6}$$

where $P$ is the predicted mask, $M$ is the ground truth, $M_e$ is the edge mask, $\odot$ denotes the point-wise product and $\lambda$ is a weighting factor balancing the two loss terms.

The edge loss applies BCE on the edge regions to emphasize boundary accuracy:

$$\mathcal{L}_{\mathrm{Edge}}(P \odot M_e, M \odot M_e) = \mathcal{L}_{\mathrm{BCE}}(P \odot M_e, M \odot M_e) \tag{7}$$

| Protocol | Method | COVERAGE | Columbia | NIST16 | CASIAv1 | IMD2020 | Average |
|----------|--------|----------|----------|--------|---------|---------|---------|
| **MVSS** | Mantra-Net (Wu et al., 2019) | 0.090 | 0.243 | 0.104 | 0.125 | 0.055 | 0.123 |
| | MVSS-Net (Chen et al., 2021) | 0.259 | 0.386 | 0.246 | 0.534 | 0.279 | 0.341 |
| | CAT-Net (Kwon et al., 2022) | 0.296 | 0.584 | 0.267 | 0.594 | 0.268 | 0.402 |
| | ObjectFormer (Wang et al., 2022) | 0.294 | 0.336 | 0.173 | 0.429 | 0.172 | 0.281 |
| | PSCC-Net (Liu et al., 2022) | 0.231 | 0.605 | 0.200 | 0.378 | 0.233 | 0.329 |
| | NCL-IML (Zhou et al., 2023) | 0.225 | 0.446 | 0.260 | 0.502 | 0.237 | 0.334 |
| | Trufor (Guillaro et al., 2023) | 0.419 | **0.865** | 0.311 | 0.721 | 0.317 | 0.527 |
| | IML-ViT (Ma et al., 2023) | 0.438 | 0.747 | 0.269 | 0.718 | 0.328 | 0.500 |
| | Mesorch (Zhu et al., 2025) | 0.276 | 0.623 | 0.283 | 0.743 | 0.256 | 0.436 |
| | SparseViT (Su et al., 2025) | 0.287 | 0.781 | 0.245 | 0.646 | 0.230 | 0.438 |
| | Relay-ViT (Ours) | 0.551 | 0.762 | **0.335** | 0.740 | **0.381** | **0.554** |
| | Relay-Seg (Ours) | **0.569** | 0.756 | 0.273 | **0.760** | 0.357 | 0.543 |
| **CAT** | Trufor (Guillaro et al., 2023) | 0.451 | 0.875 | 0.348 | 0.821 | × | 0.627 |
| | SparseViT (Su et al., 2025) | 0.513 | 0.959 | 0.384 | 0.827 | × | 0.671 |
| | Mesorch (Zhu et al., 2025) | 0.586 | 0.890 | 0.392 | **0.840** | × | 0.677 |
| | APSC-Net (Qu et al., 2024b) | 0.523 | **0.966** | 0.436 | 0.837 | × | 0.691 |
| | Relay-ViT | 0.647 | 0.878 | **0.476** | 0.806 | × | 0.702 |
| | Relay-Seg | **0.704** | 0.883 | 0.430 | 0.802 | × | **0.705** |

Table 1: Pixel-level comparison on the image manipulation localization task under both MVSS and CAT protocols. Scores indicate the F1 scores with a fixed threshold of 0.5.

## 4 EXPERIMENTS

**Datasets.** In our experiments, we conducted comprehensive evaluations using a diverse set of benchmark datasets, including CASIA v1.0 (Dong et al., 2013), CASIA v2.0 (Dong et al., 2013), Columbia (Hsu & Chang, 2006), Coverage (Wen et al., 2016), NIST16 (Guan et al., 2019), IMD2020 (Novozamsky et al., 2020), Fantastic Reality (Kniaz et al., 2019), TampCOCO (Kwon et al., 2022), DAVIS-VI (Yu et al., 2021), and MOSE100 (Lou et al., 2025). Following widely accepted and fair evaluation protocols, we adhered to the evaluation guidelines recommended by IMDLBench (Ma et al., 2024), ensuring consistency and comparability with prior studies.

**Implementation Details.** To ensure fair comparisons and consistent experimental conditions, all experiments were conducted using the IMDLBench (Ma et al., 2024) framework. We conduct experiments using ViT and SegFormer as backbones, referred to as Relay-ViT and Relay-Seg, respectively. We set the number of [GLR] tokens to $n = 2$, sub-image size to $512 \times 512$. For video, we set the sub-image size to $256 \times 256$ and the clip length to 4. We trained our models for 200 epochs using the AdamW optimizer (Loshchilov & Hutter, 2019) with a base learning rate of 1e-4, scheduled by a cosine decay policy (Loshchilov & Hutter, 2017). For more details, see the Appendix A.4.

**Evaluation Metrics.** We evaluate the performance of the predicted masks using two commonly adopted metrics: F1 score (with a fixed threshold of 0.5) and Intersection over Union (IoU).

| Methods | MOSE100 | | |
|---------|---------|---|---|
| | E2FGVI (IoU / F1) | FuseFormer (IoU / F1) | STTN (IoU / F1) |
| Mantra-Net (Wu et al., 2019) | 0.378/0.524 | 0.385/0.531 | 0.356/0.505 |
| MVSS-Net (Chen et al., 2021) | 0.038/0.057 | 0.051/0.074 | 0.094/0.133 |
| TruFor (Guillaro et al., 2023) | 0.311/0.414 | 0.285/0.388 | 0.260/0.353 |
| Focal (Yang et al., 2021) | 0.098/0.150 | 0.138/0.206 | 0.152/0.226 |
| TruVIL (Lou et al., 2024) | 0.521/0.674 | 0.557/0.699 | 0.462/0.612 |
| ViLocal (Lou et al., 2025) | 0.485/0.620 | **0.597/0.721** | 0.393/0.524 |
| Relay-ViT | 0.552/0.689 | 0.561/0.695 | **0.549/0.684** |
| Relay-Seg | **0.561/0.698** | 0.554/0.692 | 0.534/0.674 |

Table 2: Quantitative comparison on the video manipulation localization task on three different video inpainting methods. For studies without open-source implementations, we report the results as presented in their original papers to ensure a fair comparison.

| Model | Parameters (M) | GFLOPs | Note |
|-------|----------------|--------|------|
| MVSS | 150.53 | 171.01 | Input: 512×512 |
| PSCC | 3.67 | 376.83 | Input: 256×256 |
| CAT-Net | 116.74 | 137.22 | Input: 512×512 |
| TruFor | 68.70 | 236.54 | Input: 512×512 |
| Mesorch | 85.75 | 124.93 | Input: 512×512 |
| IML-ViT | 91.78 | 576.78 | Input: 1024×1024 |
| Relay-ViT | 89.55+**2.36** | 119.18 / 238.20 / 476.12 | $N = 1, 2, 4$ |
| Relay-Seg | 45.90+**2.39** | 52.71 / 105.41 / 210.83 | $N = 1, 2, 4$ |

Table 3: Model complexity comparison: parameter counts (M) and computational cost (GFLOPs). The bolded part in our models indicates additional parameters. Multiple GFLOPs values correspond to different sub-image counts $N = 1, 2, 4$.

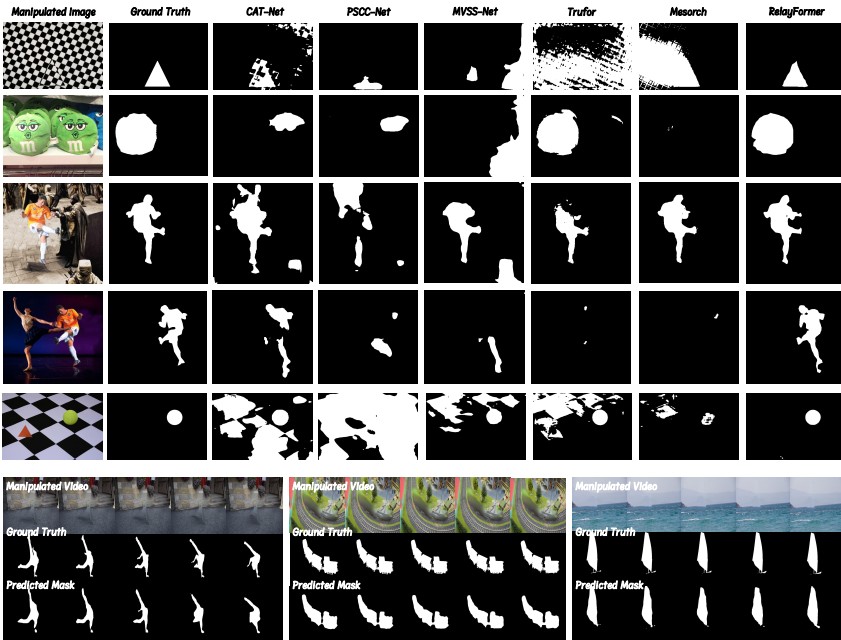

Figure 4: Visual qualitative results for image and video scenarios.

## 4.1 COMPARE WITH SoTA METHODS

**Image Manipulation Localization.** Following Protocol-MVSS (Chen et al., 2021), we train on CASIAv2 and test on others. As shown in Table 1, Relay-ViT and Relay-Seg achieve superior or competitive results across all datasets. Our framework reaches the highest average score (0.554), surpassing prior methods such as Trufor and IML-ViT. To further evaluate robustness, we also adopt Protocol-CAT. We utilize a mixed training set comprising CASIAv2, Fantastic Reality (Kniaz et al., 2019), IMD2020, and TampCOCO (Kwon et al., 2022), and evaluate on the remaining datasets (excluding IMD2020). In this challenging setting, our methods continue to excel.

**Video Manipulation Localization.** Following TruVIL (Lou et al., 2024) and ViLocal, we train on OP (Oh et al., 2019) and VI (Kim et al., 2019) edited DAVIS2016 (Perazzi et al., 2016), and test on E2FGVI (Li et al., 2022), FuseFormer (Liu et al., 2021), and STTN (Zeng et al., 2020) edited MOSE. Table 2 shows that both models achieve state-of-the-art results: Relay-Seg leads on E2FGVI, while Relay-ViT performs best on STTN, confirming robustness across different inpainting models.

As shown in Fig. 4 and in the Appendix A.9, our method also demonstrates superior performance in visual results.

| Num. | Training set | COV. | Col. | NIST16 | CASIAv1 | IMD2020 | Splice | MOSE100 |
|------|-------------|------|------|--------|---------|---------|--------|---------|
| 1 | Img+V-All | 0.569 | 0.755 | 0.282 | **0.753** | 0.357 | 0.472 | 0.684 |
| 2 | Img+V-Spl | 0.569 | 0.756 | 0.282 | 0.753 | 0.357 | **0.476** | 0.09 |
| 3 | Img+V-Inp | **0.570** | 0.733 | 0.308 | 0.748 | 0.357 | 0.133 | 0.681 |
| 4 | V-Spl | 0.051 | 0.147 | 0.163 | 0.143 | 0.219 | 0.264 | 0.119 |
| 5 | V-Inp | 0.005 | 0.139 | 0.066 | 0.029 | 0.061 | 0.003 | **0.688** |
| 6 | Img | 0.551 | **0.762** | **0.335** | 0.740 | **0.381** | 0.458 | 0.082 |

Table 4: F1 across datasets under different training configurations.

| [GLR] ($n$) | Decoder | COV. | Col. | NIST16 | CASIAv1 | IMD2020 | Average |
|-------------|---------|------|------|--------|---------|---------|---------|
| 0 | - | 0.486 | 0.596 | 0.248 | 0.691 | 0.248 | 0.454 |
| 1 | - | 0.548 | 0.718 | 0.289 | 0.751 | 0.301 | 0.521 |
| 1 | ✓ | 0.559 | 0.696 | 0.292 | 0.757 | 0.355 | 0.532 |
| 2 | ✓ | 0.551 | 0.762 | 0.335 | 0.740 | 0.381 | **0.554** |
| 3 | ✓ | 0.556 | 0.714 | 0.260 | 0.761 | 0.327 | 0.524 |

Table 5: Ablation study on manipulation detection (F1 scores) across five benchmarks. We vary the number of [GLR] tokens ($n = 0, 1, 2, 3$), ($n = 0$) means without the GLRA module, and evaluate the performance of our mask decoder.

## 4.2 INTERACTION BETWEEN IMAGE AND VIDEO IN UNIFIED TRAINING

We conduct a series of experiments to study how images and videos influence each other when trained within a unified model. Table 4 reports the F1 obtained under six training configurations: image-only (Img), video-inpainting-only (V-Inp), video-splice-only (V-Spl) (Singla et al., 2023), image + video inpainting (Img+V-Inp), and image + video inpainting + video splice (Img+V-All).

From Experiments 1, 2, and 3, we observe that adding video forgeries to image data does not noticeably improve image-domain performance. This is mainly because current video datasets lack diversity and precise annotations, while image datasets already provide rich and reliable spatial manipulation cues.

Comparing Experiments 2, 4, and 6, we find the opposite direction to be effective: image data clearly strengthen video forgery detection for manipulation types shared across both domains. High-quality image forgeries give the model a solid set of spatial cues that transfer well to video frames, effectively serving as a strong "starting point" for learning video manipulations.

Finally, Experiments 3 and 5 show that when image and video datasets contain non-overlapping manipulation types, no mutual benefit appears. Without shared artifact patterns, joint training offers no advantage over single-source training.

## 4.3 FLOPS AND PARAMETERS

As shown in Table 3, our framework adapts dynamically to varying input resolutions, reducing redundant computation with minimal parameter overhead. See the Appendix A.8 for a more detailed analysis of time complexity and parallelism.

## 4.4 ABLATION STUDY

We conduct ablation experiments to assess the contribution of each component from three perspectives: (1) the number of [GLR] tokens and the role of the GLRA module ($n=0$), (2) the Query-based Mask Decoder, and (3) spatial-temporal cues and interpolation strategies.

Table 5 reports results on five benchmarks. Adding a single [GLR] token ($n=1$) improves results, and substituting the MLP with our decoder further boosts performance (0.532). The best performance occurs at $n=2$, while $n=3$ slightly degrades results due to redundancy (Fig. 6). We provide

| Spatial | Temporal | F1 | IoU |
|:---:|:---:|:---:|:---:|
| – | – | 0.6124 | 0.4828 |
| ✓ | – | 0.6745 | 0.5391 |
| ✓ | ✓ | **0.6877** | **0.5524** |

Table 6: Ablation of GLRA along spatial and temporal dimensions (MOSE100).

| Metric | w/o resize | w/ resize |
|:---|:---:|:---:|
| Res. | 2958×4437 | 1024×1024 |
| F1 | **0.453** | 0.350 |

Table 7: Impact of interpolation (IMD2020). *Res.* denotes the maximum resolution.

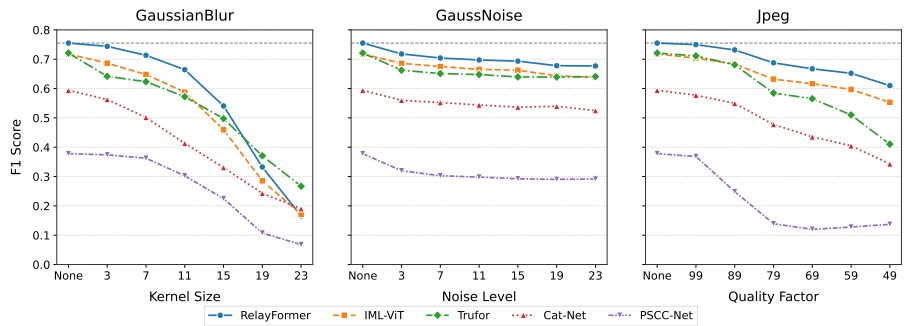

Figure 5: Robustness analysis results of the model under common perturbations.

a further analysis of the behavior of the `[GLR]` token along with additional visualizations in the Appendix A.5.

**Effectiveness of GLRA along Temporal Dimensions** We further investigate the effectiveness of applying GLRA solely along the spatial dimension versus jointly across both spatial and temporal dimensions in video detection, in order to verify that our method can indeed extend to capturing temporal information in videos. The corresponding results are presented in Table 6.

**Effect of Input Resolution on Performance and Extrapolation to Higher Resolutions** Table 7 illustrates the impact of input resolution on detection accuracy. Resizing 4K inputs to 1024×1024 substantially degrades performance (0.350 vs. 0.453), as interpolation obscures high-frequency forensic artifacts. Conversely, the model demonstrates robust extrapolation capability. Despite being trained on lower resolutions (max 600×901), it achieves peak performance on raw 4K images. This confirms that the model generalizes beyond its training scale, effectively leveraging the fine-grained cues available at higher resolutions.

## 4.5 ROBUSTNESS EVALUATION

We assess the robustness of different methods under common corruptions: Gaussian Blur, Gaussian Noise, and JPEG Compression. As shown in Fig. 5, RelayFormer consistently outperforms prior methods across all distortion types and levels. It maintains higher F1 scores under increasing blur, noise, and compression, demonstrating strong generalization to real-world degradations.

## 5 CONCLUSION

In this work, we introduce **RelayFormer**, a unified framework tackling resolution diversity and temporal extension in manipulation localization. By employing Global Local Relay (GLR) tokens across fixed-size sub-images, our relay-based attention efficiently propagates scene-level context while preserving fine-grained forensic evidence. This design enables the processing of variable resolutions and video sequences. Extensive experiments demonstrate that RelayFormer achieves superior performance and computational efficiency, offering a scalable solution for both static and temporal visual data.

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

# A APPENDIX

## A.1 LIMITATIONS

**Limitations on global context modeling.** While our partition-and-relay design achieves a favorable trade-off between accuracy and efficiency, it inevitably introduces a limitation compared to full global attention. Specifically, when manipulations span across multiple sub-images, the Global-Local Relay Attention (GLRA) propagates contextual information through relay tokens rather than establishing exhaustive pairwise interactions. This relay mechanism is computationally more efficient, yet it cannot capture cross-partition dependencies as precisely as a global attention scheme if computational constraints are disregarded.

## A.2 DETAILED DESCRIPTION OF THE METHOD

### A.2.1 INPUT UNIFICATION

To more clearly demonstrate how we preprocess videos and images into a unified form, we provide relevant pseudocode 1.

---

**Algorithm 1** Sub-images Extraction

---

**Require:** Image set $\mathcal{X}_{\text{img}}$, video set $\mathcal{X}_{\text{vid}}$, patch size $(H_p, W_p)$, stride $(S_h, S_w)$
**Ensure:** Unified tensor $\mathbf{X} \in \mathbb{R}^{B_{\text{total}} \times C \times H_p \times W_p}$
 1: $\mathbf{X} \leftarrow$ empty list
 2: **for each** image $\mathbf{x} \in \mathcal{X}_{\text{img}}$ **do**
 3:      $H, W \leftarrow$ spatial dimensions of $\mathbf{x}$
 4:      $N_h \leftarrow \lceil (H - H_p)/S_h \rceil + 1$
 5:      $N_w \leftarrow \lceil (W - W_p)/S_w \rceil + 1$
 6:      Extract $N_h \times N_w$ patches using sliding window
 7:      Append patches to $\mathbf{X}$
 8: **end for**
 9: **for each** video $\mathbf{x} \in \mathcal{X}_{\text{vid}}$ **do**
10:      $T, H, W \leftarrow$ dimensions of $\mathbf{x}$
11:      $N_h \leftarrow \lceil (H - H_p)/S_h \rceil + 1$
12:      $N_w \leftarrow \lceil (W - W_p)/S_w \rceil + 1$
13:      Extract patches from first frame: $\mathbf{P} \leftarrow$ patches from $\mathbf{x}[0]$
14:      Repeat $\mathbf{P}$ along temporal dimension: $\mathbf{P}_{\text{full}} \leftarrow \text{repeat}(\mathbf{P}, T)$
15:      Append $\mathbf{P}_{\text{full}}$ to $\mathbf{X}$
16: **end for**
17: Stack all patches into tensor of shape $(B_{\text{total}}, C, H_p, W_p)$
18: **return** $\mathbf{X}$

---

### A.2.2 PARAMETER-EFFICIENT STRATEGY

Using shared parameters for both local and global attention severely degrades performance because the two modules serve fundamentally different purposes and operate on distinct feature spaces. Local attention works on dense, low-level patch tokens ($X_i$), focusing on fine-grained textures, edges, and object parts. Global attention, by contrast, processes sparse, high-level `[GLR]` tokens ($T_{\text{flat}}$), which summarize sub-images and model long-range dependencies. A single set of projection weights cannot simultaneously specialize in local detail extraction and global structural reasoning, leading to suboptimal representations in both tasks. Therefore, separate parameterization is essential to preserve both local fidelity and global coherence.

As shown in Fig. 3, our solution introduces functional specialization via a dynamic parameter-sharing scheme based on Low-Rank Adaptation (LoRA). Each Transformer block maintains a shared set of backbone projection matrices ($W_Q$, $W_K$, $W_V$), which are fully trainable. On top of this backbone, we add two task-specific sets of LoRA parameters: $\{A_{\text{local}}, B_{\text{local}}\}$ for SelfAttn$_{\text{local}}$ and $\{A_{\text{global}}, B_{\text{global}}\}$ for SelfAttn$_{\text{global}}$. During the forward pass, the effective weight is dynamically

constructed. For example, in local attention:

$$W_Q' = W_Q + B_{Q,\text{local}} A_{Q,\text{local}},$$

while in global attention:

$$W_Q'' = W_Q + B_{Q,\text{global}} A_{Q,\text{global}}.$$

Here, $W_Q$ provides a shared backbone, and LoRA contributes lightweight, context-specific adjustments.

Unlike conventional LoRA fine-tuning, our backbone remains trainable, and the LoRA parameters are never merged into it. This design is crucial: rather than adapting a frozen model to a single task, we enable two co-existing functional modes that can be switched dynamically. The result is efficient parameter sharing that preserves specialization for both local and global reasoning.

## A.3 DATASETS

### A.3.1 IMAGE DATASETS

We use the following publicly available datasets for the detection of spliced and copy-moved images, following previous settings (Ma et al., 2023; 2024), we didn't use real images in all datasets:

To provide a detailed overview of these datasets, Table 8 summarizes key attributes, including the number of images or videos, forgery types, and other relevant characteristics. All details are sourced from official or authoritative descriptions to ensure reliability.

Table 8: Overview of benchmark datasets for image forgery detection. Forgery types include splicing (S), copy-move (C), removal/inpainting (R), and others (O).

| Dataset | Year | # Authentic | # Forged |
|---|---|---|---|
| **Image Forgery Datasets** | | | |
| CASIA v1.0 | 2013 | 800 | 921 |
| CASIA v2.0 | 2013 | 7,491 | 5,123 |
| Columbia | 2004 | 183 | 180 |
| Coverage | 2016 | 100 | 100 |
| NIST16 | 2016 | 560 | 564 |
| IMD2020 | 2020 | 414 | 2010 |
| **AI-Generated Forgery Datasets** | | | |
| AutoSplice | 2023 | 2,273 | 3,621 |
| CocoGlide | 2023 | - | - |

### A.3.2 VIDEO DATASETS

For video inpainting experiments, we use the following datasets:

| Dataset | Use |
|---|---|
| DAVIS 2016 (Perazzi et al., 2016) | Generate training |
| MOSE (Ding et al., 2023) | Cross-dataset test |

Table 9: Video datasets used in our workflow.

**Details of usage.** DAVIS 2016 contains 50 short videos, split into 30 training and 20 validation clips (Perazzi et al., 2016). We use two video inpainting models—OP (Oh et al., 2019) and VI (Kim et al., 2019)—to generate corrupted–reconstructed frame pairs for training.

The MOSE dataset includes videos and object masks with complex scenarios (Ding et al., 2023). We use the validation split of 100 clips as a test set for evaluating, using three methods: E2FGVI (Li et al., 2022), FuseFormer (Liu et al., 2021), and STTN (Zeng et al., 2020) models to create validation datasets.

### A.3.3 Data Split Summary

- **Image tasks:** CASIA v2.0 is used for training. Other image datasets (CASIA v1.0, Columbia, Coverage, NIST16, IMD2020) are used for cross-dataset testing.

- **Video tasks:** DAVIS 2016 is used to generate training data via OP and VI models. MOSE validation split is used for testing with E2FGVI, FuseFormer, and STTN.

### A.4 Implementation Details

To ensure fairness and consistency, all experiments are conducted with the IMDLBench (Ma et al., 2024) framework and follow the training configuration of IML-ViT. ViT and SegFormer backbones are adopted (denoted as Relay-ViT and Relay-Seg). All Transformer blocks are replaced by GLRA modules, and the number of [GLR] tokens is set to $n = 2$. For 4D RoPE, following (Wang et al., 2024), each of the four structured axes ($T$, $id$, $H$, $W$) is assigned an equal dimension size, and any leftover dimensions are grouped into the remaining part.

| Component | Setting |
|---|---|
| Backbones | ViT-Base-patch16, SegFormer |
| [GLR] tokens | $n = 2$ |
| Sub-image size (image) | $528 \times 528$ |
| Sub-image size (video) | $256 \times 256$ |
| Temporal clip length | 4 frames |
| Mask decoder layers ($K$) | 3 |
| Learnable queries | 8 |
| Frozen epoch | 1 (freeze pre-trained weights) |
| Edge loss weight $\lambda$ | 20 |
| LoRA rank | 8 |
| LoRA scaling factor | 2 |

Table 10: Model and Architecture Settings.

**Image and Video Preprocessing.** Images larger than $1024 \times 1024$ are resized by scaling the longer side to 1024 while maintaining aspect ratio, followed by zero-padding to $1024 \times 1024$. For video data, we apply a similar process.

**Data Augmentation.** We follow IML-ViT (Ma et al., 2023) and apply:

- re-scaling and random horizontal flipping,

- Gaussian blurring and random rotation,

- copy-move and inpainting of rectangular regions.

### A.5 Understanding GLRA Behavior

To further investigate the effect of GLRA, we visualize intermediate attention maps and feature activation patterns in Fig. 6 and Fig. 7. Without GLRA, the representations of different spatial sub-images tend to diverge, as local self-attention lacks a mechanism for sufficient global interaction. This leads to fragmented feature distributions that fail to capture cross-region dependencies. In contrast, GLRA introduces an explicit relay pathway for long-range communication, enforcing semantic consistency across sub-images.

Moreover, we find that when employing two [GLR] tokens, each token naturally specializes in distinct spatial regions, suggesting a clear division of labor that supports complementary global reasoning. However, when the number of relay tokens increases to three, one of the tokens consistently collapses into an attention pattern nearly identical to that of another token, indicating redundancy and an unclear functional role, as shown at the bottom of Fig. 6. We hypothesize that this redundancy introduces competition among relay tokens, weakening their ability to form stable and

| Parameter | Value |
|---|---|
| GPUs | $4 \times$ NVIDIA RTX 3090 |
| Precision | AMP (mixed precision) |
| Per-GPU batch size | 4 |
| Gradient accumulation steps | 4 |
| Effective batch size | 64 |
| Epochs | 200 |
| Optimizer | AdamW (Loshchilov & Hutter, 2019) |
| Base learning rate | $1 \times 10^{-4}$ |
| LR schedule | Cosine decay (Loshchilov & Hutter, 2017) |
| Warmup epochs | 2 |
| Minimum learning rate | $5 \times 10^{-7}$ |
| Weight decay | 0.05 |
| Random seed | 42 |
| Test-time augmentation | None |

Table 11: Training Configuration.

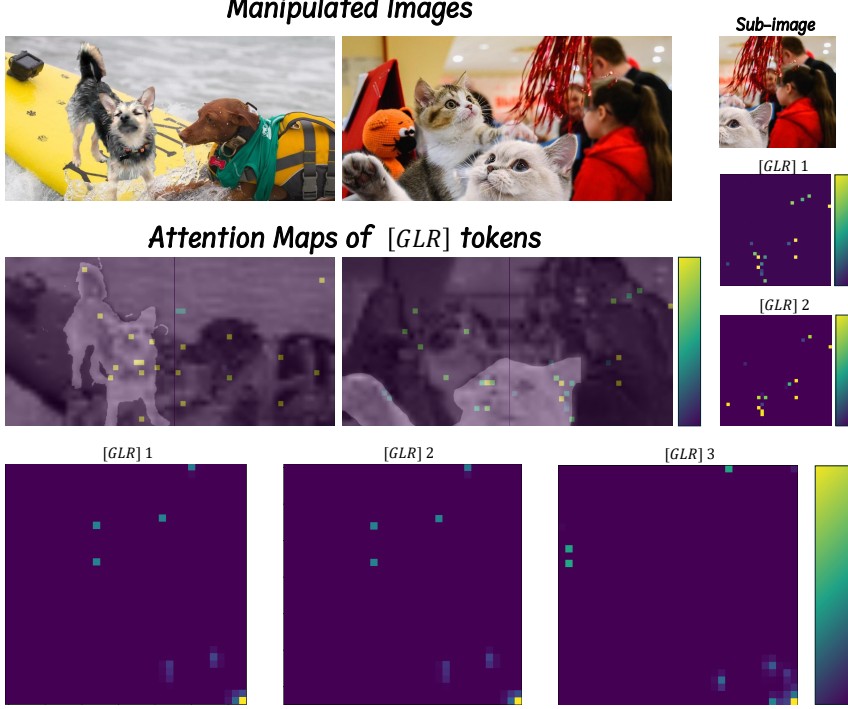

Figure 6: Attention map visualization of [GLR] tokens for analyzing their behavioral patterns.

meaningful global interactions. Consequently, the representational ambiguity introduced by the extra token leads to diminished performance. These observations support our empirical finding that setting the number of relay tokens to $n = 2$ strikes an effective balance between modeling capacity and computational efficiency.

## A.6    ADDITIONAL CROSS-DATASET EVALUATION

We further evaluate our models on three additional datasets, with results summarized in Table 12. Overall, our Relay-based methods consistently achieve leading performance under broad testing conditions. In particular, Relay-Seg attains the highest average F1 score (0.372), surpassing all com-

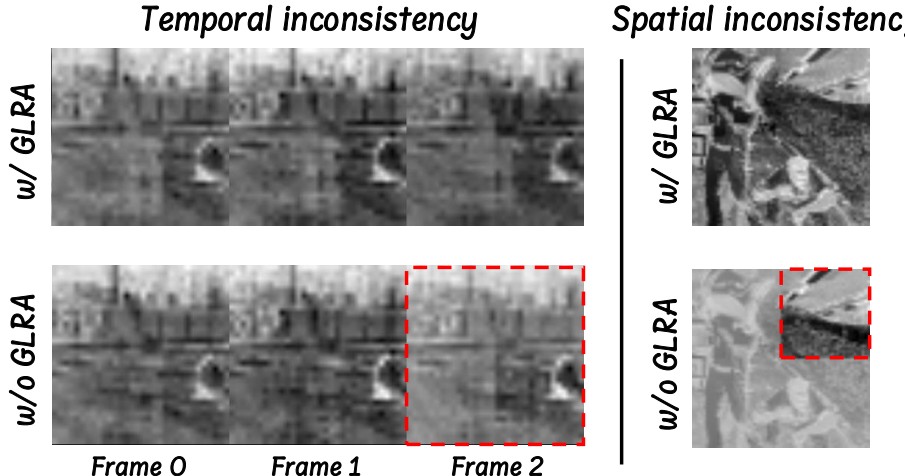

Figure 7: Qualitative results illustrating the role of GLRA.

peting baselines. This demonstrates that the proposed GLRA mechanism generalizes well across diverse forgery types and input distributions.

A noteworthy observation is the impact of backbone choice when facing AI-generated forgery datasets. For example, both Relay-ViT and IML-ViT, which are based on Vision Transformer architectures, achieve a similar level of performance. In contrast, Relay-Seg outperforms other methods, while SparseViT also exhibits competitive results.

These findings not only validate the robustness of our relay-based formulation but also suggest that backbone-level inductive biases play a significant role in detecting AI-generated content. Importantly, our approach benefits from these architectural strengths while introducing only minimal overhead, thereby offering both efficiency and adaptability.

| Method | AutoSplice | CocoGlide | Defacto-12k | Average |
|---|---|---|---|---|
| IML-ViT (Ma et al., 2023) | 0.221 | 0.210 | 0.367 | 0.266 |
| SparseViT (Su et al., 2025) | **0.386** | 0.142 | 0.242 | 0.257 |
| Mesorch (Zhu et al., 2025) | 0.216 | 0.120 | 0.292 | 0.209 |
| Relay-ViT | 0.289 | 0.203 | **0.416** | 0.303 |
| Relay-Seg | 0.379 | **0.328** | 0.409 | **0.372** |

Table 12: Performance comparison of different methods on three datasets (metric: F1 score, higher is better)

A.7 MORE ABLATION STUDIES

**Studies on the Model Architecture.** To evaluate the effectiveness of GLRA when integrated into different parts of the backbone, we conducted experiments with three replacement strategies on the CASIA v1 dataset: (1) inserting GLRA into a sparse set of layers across the transformer encoder $[0, 3, 7, 11]$, (2) replacing all layers in the latter half of the encoder $[6–11]$, and (3) replacing all layers in the encoder (i.e., full replacement). As shown in Figure 8, progressively increasing the number of GLRA-applied layers leads to consistent improvements in F1 scores: 73.24%, 74.21%, and 75.50%, respectively. These results indicate that GLRA contributes more significantly when applied to deeper layers and that full-layer integration yields the best performance. This suggests that GLRA is both effective and scalable when applied throughout the model architecture.

**Effect of 4D RoPE.** To isolate the impact of positional encoding, we remove the 4D RoPE from the GLRA module and observe both training stability and final performance. As shown in Table 13,

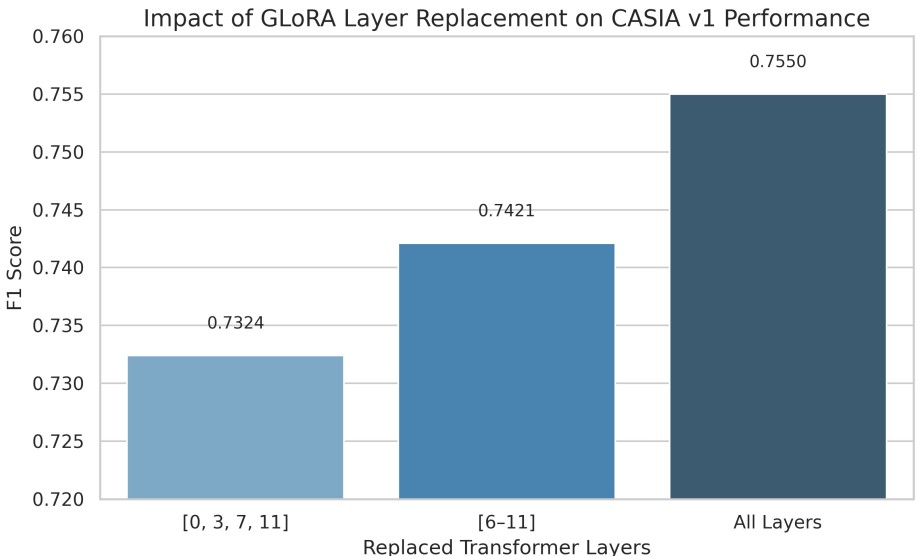

Figure 8: Impact of GLRA layer replacement strategies on CASIA v1.

the removal of RoPE leads to significantly more unstable training dynamics, exhibiting larger gradient fluctuations and slower convergence. Performance is also consistently worse across all datasets, with the average score dropping from 0.554 to 0.517. These findings indicate that 4D RoPE provides essential spatial consistency for the compressed relay representations, stabilizing the optimization process and enhancing the model's ability to capture long-range structural information.

| RoPE | Coverage | Columnbia | NIST | CASIAv1 | Avg. |
|---|---|---|---|---|---|
| × | 0.542 | 0.682 | 0.301 | 0.707 | 0.517 |
| ✓ | **0.551** | **0.762** | **0.335** | **0.740** | **0.554** |

Table 13: Ablation on the use of 4D RoPE in the GLRA module. Removing RoPE results in unstable training and degraded performance.

**Ablation on Sub-Image Size and Overlap.** For video inputs, we adopt a sub-image size of $256 \times 256$. The primary constraint is the resolution of the training data, whose maximum spatial size is $512 \times 512$. Using $512 \times 512$ sub-images would reduce GLRA to full attention. Hence, $256 \times 256$ is the largest feasible choice that preserves spatial hierarchy while maintaining the intended behavior of GLRA.

For image inputs, larger sub-images provide stronger discriminative capacity but incur higher quadratic computational cost. We adopt $512 \times 512$ as a balanced choice between accuracy and efficiency. To further illustrate this trade-off, we additionally compare $256 \times 256$ sub-images with and without overlap. The results are summarized in Table 14.

| Coverage | Columnbia | NIST | CASIA | IMD2020 | Avg. | overlap | size |
|---|---|---|---|---|---|---|---|
| 0.389 | 0.658 | 0.260 | 0.685 | 0.355 | 0.469 | w/o | 256 |
| 0.434 | 0.645 | 0.261 | 0.709 | 0.337 | 0.477 | w/ | 256 |
| 0.551 | 0.762 | 0.335 | 0.740 | 0.381 | 0.554 | w/ | 512 |

Table 14: Ablation on sub-image size and overlap for image inputs.

These results show that $512 \times 512$ yields notable performance gains over the $256 \times 256$ setting, supporting our choice as an effective trade-off between accuracy and computational cost.

We use an overlap of 16 pixels, corresponding to exactly one ViT-Base patch. This ensures continuity across adjacent sub-images while remaining compatible with the pretrained backbone. Larger overlaps substantially increase computation with limited benefit, whereas removing overlap weakens cross-region consistency.

## A.8 COMPLEXITY AND PARALLELISM ANALYSIS

As shown in Table 3, both Relay-ViT and Relay-Seg introduce only a negligible number of additional parameters ($\sim$ 2.4M) compared to their respective backbones, demonstrating that GLRA incurs minimal overhead. Despite this, our methods substantially reduce computational cost relative to prior transformer-based baselines. For example, Relay-ViT achieves a lower GFLOPs budget than IML-ViT even when operating with $N=4$ sub-images at $1024 \times 1024$ resolution. Moreover, the scalability of our design is evident: the GFLOPs grow linearly with the number of sub-images, while the parameter count remains nearly constant. This highlights the efficiency of our relay-based formulation, which decouples global reasoning capacity from the quadratic growth in input resolution. Overall, Relay-ViT and Relay-Seg strike a favorable balance between model size, computational efficiency, and representational power, validating the practical advantage of the proposed GLRA mechanism.

**Time complexity.**  In the local attention stage, each sub-image $U_i$ contains $P$ patch tokens and $m$ [GLR] tokens, yielding a total of $P + m$ tokens per sub-image. The self-attention operation in Eq. equation 1 thus requires $\mathcal{O}((P+m)^2 d)$ computations per layer, where $d$ is the hidden dimension. Across all sub-images in the batch, the total complexity scales linearly with $B_{\text{total}}$, the unified number of sub-images.

In the global attention stage, the complexity depends only on the number of [GLR] tokens. For a sample with $N_i$ sub-images, the concatenated sequence length is $N_i \cdot m$, leading to a cost of $\mathcal{O}((N_i m)^2 d)$ per layer in Eq. equation 3. Compared to local attention, this is relatively lightweight, since $m \ll P$ and $N_i$ is typically small.

**Parallelism.**  The unified representation in Sec. 3.1 enables straightforward parallelization across all sub-images. However, the number of sub-images $N_i$ may vary across samples due to differing input resolutions, resulting in variable numbers of [GLR] tokens. To enable efficient batched computation, we pad the sequence of [GLR] tokens in each sample to the maximum length within the batch. This ensures that global attention can be executed in parallel without irregular memory access patterns. Since the number of sub-images per sample is usually small, the overhead introduced by such padding is negligible in practice, while the benefit of full parallelization is substantial.

## A.9 MORE VISUALIZATION RESULTS

In this section, we provide additional qualitative comparisons to further demonstrate the effectiveness of our method. Figure 9 showcases a variety of manipulated images and videos, along with their corresponding ground truth masks and the predicted results from different baseline methods, including CAT-Net, PSCC-Net, Trufor, and Mesorch. Our method, RelayFormer, consistently generates more accurate and fine-grained manipulation masks, with better localization and fewer false positives compared to previous approaches.

For manipulated video sequences, our model not only detects spatial tampering more precisely but also captures temporal consistency across frames, which is essential for robust manipulation detection in videos.

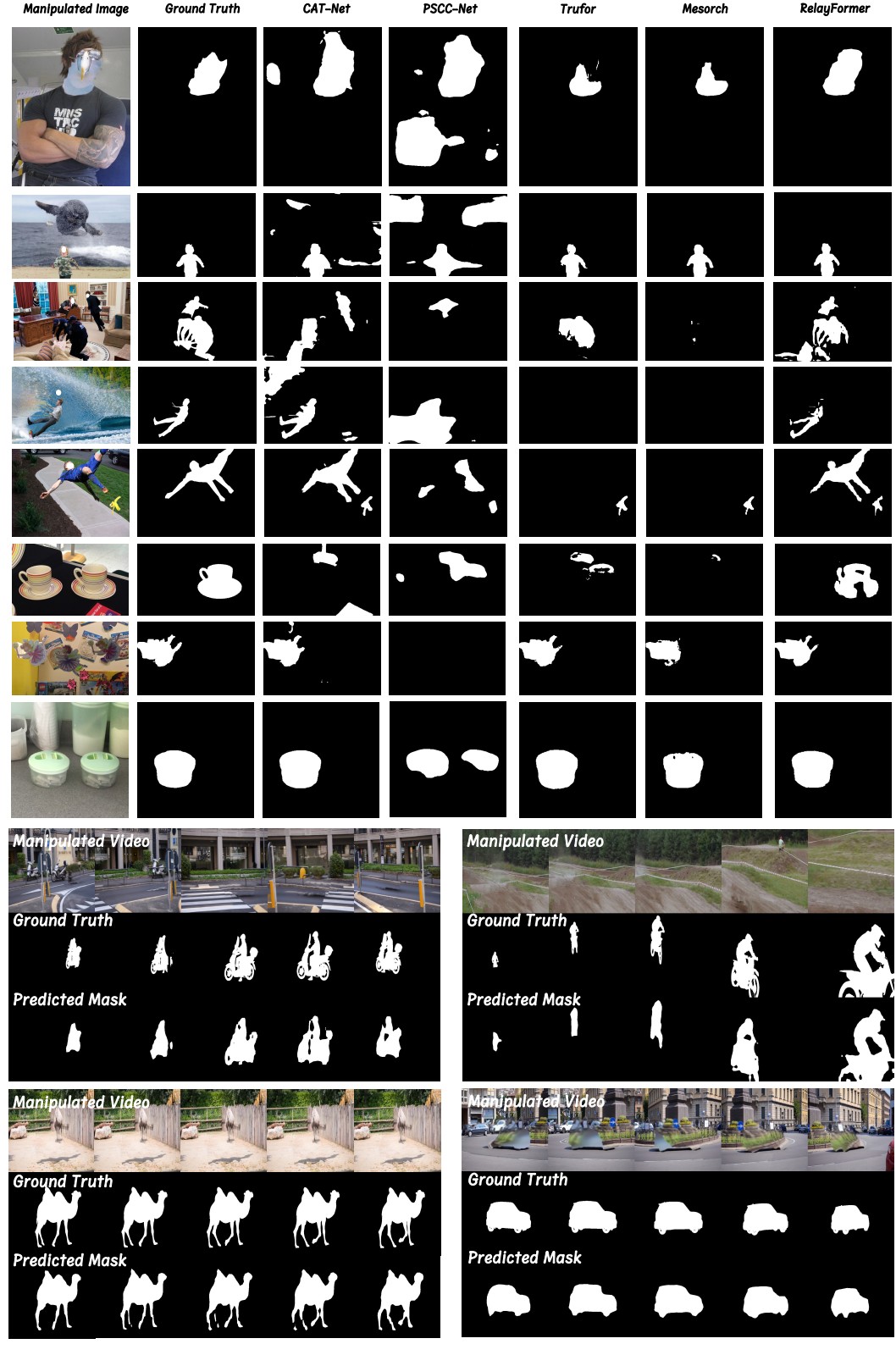

Figure 9: Qualitative comparisons on manipulated images and videos. Our method (RelayFormer) shows superior performance in both spatial and temporal prediction accuracy compared to prior methods.

## B  STATEMENT ON THE USE OF LARGE LANGUAGE MODELS (LLMs)

In the preparation of this manuscript, a Large Language Model (LLM) was used solely for the purpose of language polishing, including minor grammar correction and stylistic refinement of the authors' original text. The LLM did not contribute to the conceptualization of the research, the design of experiments, the analysis of results, or the interpretation of findings. All research ideas, methods, and conclusions presented in this paper are entirely the work of the authors.

