# OpenReview forum: "RelayFormer: A Unified Local-Global Attention Framework for Scalable Image and Video Manipulation Localization"
_ICLR.cc/2026/Conference — ICLR 2026 Poster_

### Official Review · Reviewer_k3yb · 2025-10-27

**Soundness:** 3
**Presentation:** 2
**Contribution:** 3
**Rating:** 6
**Confidence:** 5

**Summary:**

This paper introduces RelayFormer, a unified attention framework designed to address two core challenges in Visual Manipulation Localization (VML): resolution diversity and the modality gap across images and videos. The core of RelayFormer is a novel Global-Local Relay Attention (GLRA) mechanism. This mechanism partitions inputs of arbitrary resolution into fixed-size sub-images and introduces a small set of learnable "relay" tokens to absorb fine-grained manipulation artifacts locally before engaging in a sparse information exchange at the global level to propagate context. Furthermore, the framework incorporates a parameter-efficient design, leveraging a shared Transformer backbone with task-specific LoRA adapters to optimize efficiency with minimal impact on performance. The authors validate the framework's efficiency and performance through extensive experiments on multiple image and video benchmarks.

**Strengths:**

1. The core GLRA mechanism introduces a small set of learnable [GLR] (Global-Local Relay) tokens to act as information proxies. This design reduces the computational complexity of global attention from quadratic in the total number of patches to quadratic in the number of sub-images, achieving an exponential reduction in computational cost. The framework employs an elegant information fusion strategy, iteratively executing a two-step process of "local perception" and "global relay" within each Transformer layer.

2. The paper ingeniously employs a strategy that combines a shared Transformer backbone with two independent, lightweight LoRA adapters. By dynamically switching between these adapters, the design achieves functional specialization for the distinct local and global attention tasks at a minimal additional parameter cost.

**Weaknesses:**

1. Information Bottleneck of Relay Tokens:
Since global information exchange is mediated through a small set of relay tokens, this mechanism may introduce mild information loss when manipulations span multiple sub-images with complex cross-region dependencies (e.g., fine-grained copy-move forgeries).
Although this trade-off is computationally justified, it would be useful to quantify how much contextual precision is sacrificed compared to a full-attention baseline.


2. Missing CAT Protocol Comparison:
The image manipulation localization experiments primarily follow the MVSS protocol (training on CASIA-v2 and testing on standard benchmarks).
However, stronger and more comprehensive evaluation settings, such as the CAT protocol,should also be considered.

**Questions:**

1. Quantifying Relay Compression Loss:
How much performance degradation, if any, occurs when comparing GLRA against a full global-attention model (even at a reduced resolution)?
Such an experiment could empirically validate the trade-off between efficiency and precision.

2. Evaluation under CAT Protocol:
Given that the paper currently reports results under the MVSS protocol, could the authors also train and evaluate a CAT-protocol version of RelayFormer?

3. Temporal Sparsity in Manipulation Frames
In real-world video forgeries, only a few frames may be manipulated while the majority remain authentic.
Given that RelayFormer aggregates temporal information through relay tokens across frames, how effectively can the model detect and localize sparse manipulations that occur in only a small subset of frames?

---

> ### Author Response · Authors · 2025-11-21
> **Response to Reviewer k3yb (Part 1)**
>
> > **Q1: Quantized GLRA Compression Loss**
>
> **A1:** We are grateful for the reviewer’s constructive suggestion to quantify the "relay compression loss." We agree that analyzing the trade-off between the information bottleneck and computational efficiency is crucial.
>
> **Quantifying Relay Compression Loss (GLRA vs. Full Attention).**
> To address your concern regarding potential information loss mediated by relay tokens, we performed a comparative experiment against a **Full Self-Attention** baseline. To make the full-attention baseline computationally feasible, we operated at a reduced input resolution.
>
> **Experimental Results:**
> The comparison results are summarized in **Table 1** below.
>
> Table 1 Quantized GLRA Compression Loss.
>
> | Method                    | Coverage | Columbia |  NIST  | CASIAv1 |  Avg.  |
> | :------------------------ | :------: | :------: | :----: | :-----: | :----: |
> | **Self-Attention (Full)** |  0.5690  |  0.9012  | 0.4711 | 0.8158  | 0.6893 |
> | **GLRA (Ours)**           |  0.4977  |  0.8908  | 0.4575 | 0.8442  | 0.6726 |
>
> **Analysis:**
>
> 1.  **Minimal Performance Degradation:** The results indicate that the information bottleneck introduced by GLRA leads to only a marginal drop in average accuracy (**0.6726** for GLRA vs. **0.6893** for Full Attention). This confirms your hypothesis that the loss is "mild."
> 2.  **Conclusion:** These empirical results validate the design of GLRA. The sparse global information propagation is sufficient for capturing scene-level regularities (such as illumination patterns and object semantics) while offering a significant advantage in computational efficiency.

---

> ### Author Response · Authors · 2025-11-21
> **Response to Reviewer k3yb (Part 2)**
>
> > **Q2: Evaluation under CAT Protocol**
>
>
> **A2:** We thank the reviewer for the suggestion. Following the reviewer’s advice, we have additionally trained and evaluated RelayFormer under the official CAT protocol. As shown in the updated Table 1 in the revised manuscript, both Relay-ViT and Relay-Seg outperform previous methods, and Relay-Seg achieves the best results. These results will be added to the camera-ready version along with an updated comparison table.
>
> | Method        | COVERAGE  | Columbia  | NIST16    | CASIAv1   | Average   |
> | ------------- | --------- | --------- | --------- | --------- | --------- |
> | Trufor        | 0.451     | 0.875     | 0.348     | 0.821     | 0.627     |
> | SparseViT     | 0.513     | **0.959** | 0.384     | 0.827     | 0.671     |
> | Mesorch       | 0.586     | 0.890     | 0.392     | **0.840** | 0.677     |
> | APSC-Net      | 0.523     | **0.966** | **0.436** | 0.837     | 0.691     |
> | **Relay-ViT** | **0.647** | 0.878     | **0.476** | 0.806     | **0.702** |
> | **Relay-Seg** | **0.704** | 0.883     | 0.430     | 0.802     | **0.705** |
>
>
> > **Q3: Temporal Sparsity in Manipulation Frames**
>
> **A3:** We thank the reviewer for this insightful question regarding temporal sparsity. We acknowledge that in real-world scenarios, manipulations are sometimes sparse and discontinuous.
>
> Our RelayFormer is capable of effectively detecting such sparse manipulations for the following reasons:
>
> 1.  **Strong Spatial Foundation:** The architecture is designed to extract robust features from individual frames first. The temporal modeling (via Relay tokens) serves as an enhancement module rather than a bottleneck. Even without distinct temporal inconsistencies, the model preserves strong sensitivity to spatial artifacts within single frames.
>
> 2.  **Complementary Gains:** As validated by our ablation study, the model can effectively rely on spatial cues when temporal clues are weak or sparse. When manipulations are continuous, the temporal information further boosts performance.
>
> Table A. Ablation of GLRA along spatial and temporal dimensions.
> | Spatial | Temporal | F1         | IoU        |
> | ------- | -------- | ---------- | ---------- |
> | ✓       | --       | 0.6745     | 0.5391     |
> | ✓       | ✓        | **0.6877** | **0.5524** |
>
> To demonstrate this, we refer to the ablation study in **Table 6** of our paper (Table A. above). The results show that the "Spatial-only" variant already achieves competitive performance. Introducing temporal interaction yields further improvements, confirming that the temporal module enhances detection without compromising the model's ability to detect sparse, frame-level forgeries.

---

### Official Review · Reviewer_M9mm · 2025-10-29

**Soundness:** 3
**Presentation:** 3
**Contribution:** 3
**Rating:** 4
**Confidence:** 4

**Summary:**

This paper proposes the RelayFormer framework for unified visual manipulation localization in both images and videos. The authors claim their method addresses two primary challenges: resolution diversity and the image-video "modality gap." The core innovation is the Global-Local Relay Attention (GLRA) mechanism, which propagates global context across fixed-size sub-images through learnable Global-Local Relay (GLR) tokens. The paper divides inputs into overlapping sub-images to avoid interpolation, introduces GLR tokens as an information bottleneck for efficient global-local interaction, and employs a LoRA-based parameter-efficient strategy. Experimental validation is conducted on multiple image and video manipulation datasets.

**Strengths:**

1.The research problem is well-motivated. The paper correctly identifies that manipulation detection requires simultaneous consideration of both fine-grained local artifacts and coarse-grained global consistency cues (such as illumination mismatches and structural redundancies). This provides a sound foundation for proposing an efficient global information propagation mechanism and offers valuable insights for the forensics community.

2.The input unification strategy (Section 3.1) is practical. By partitioning variable-resolution images and videos into fixed-size overlapping sub-images, the method avoids interpolation operations that could destroy forensic traces. This strategy is both simple and effective, supports parallel processing, and demonstrates practical engineering value.

3.The experimental validation is comprehensive. The paper evaluates the approach on 8 datasets covering both image and video scenarios. Ablation studies systematically analyze GLR token quantity (Table 4), temporal dimension (Table 5), and interpolation effects (Table 6). Robustness evaluation (Figure 5) tests against common perturbations, with generally sound experimental design.

**Weaknesses:**

1. The authors emphasize that images and videos belong to different modalities, which is imprecise. Multimodality typically refers to different data types (e.g., vision and text, vision and audio). Both images and videos belong to the visual modality, differing only in the presence or absence of temporal dimension. This conceptual confusion weakens the theoretical positioning of the paper. It would be more appropriate to frame this as a "temporal dimension extension problem" rather than a "modality gap."

2. The technical novelty is limited. The technical contributions of this work are relatively incremental. The use of fixed-size sub-images and Global-Local Relay (GLR) tokens represents fairly standard techniques. Fixed-size sub-images are a common operation in image and video processing (standard practice in ViT and similar architectures). The proposed GLR tokens show no significant distinction from existing prompt tuning methods [1][2][3][4], as they essentially all perform information aggregation through learnable tokens. The paper fails to adequately differentiate GLR tokens from prior work such as Dgl [3] and Visual Prompt Tuning [4]. The innovative contributions are primarily at the application level rather than the methodological level.
[1] Liu, Xiao, et al. "P-tuning v2: Prompt tuning can be comparable to fine-tuning universally across scales and tasks." arXiv preprint arXiv:2110.07602 (2021).
[2]Zhang, Ji, et al. "Dept: Decoupled prompt tuning." Proceedings of the IEEE/CVF Conference on Computer Vision and Pattern Recognition. 2024.
[3] Yang, Xiangpeng, et al. "Dgl: Dynamic global-local prompt tuning for text-video retrieval." Proceedings of the AAAI Conference on Artificial Intelligence. Vol. 38. No. 7. 2024.
[4] Jia, Menglin, et al. "Visual prompt tuning." European conference on computer vision. Cham: Springer Nature Switzerland, 2022.

3. The readability needs improvement and the related work section is weak. The writing quality requires enhancement. For example, lines 103-104 and 152-153 employ excessive use of dashes. The Related Work section (Section 2) is overly brief, appearing more as a literature listing than a critical analysis. It lacks in-depth discussion of why existing methods fail. The writing should be more direct and professional to enhance clarity.

4. The 4D RoPE design lacks sufficient justification. The 4D RoPE positional encoding design (Section 3.2) lacks adequate theoretical foundation. How are the dimension allocations (d_T, d_id, d_H, d_W, d_rem) determined (lines 261)? The paper only provides formulas without sensitivity analysis. More importantly, the paper claims "strong extrapolation capability" (line 270) but provides no experimental verification. For instance, what is the performance on 2K or 4K images after training at 512×512 resolution? This claim lacks empirical support.

5. Hyperparameter selection lacks rigor. The choice of n=2 for GLR token quantity (line 311) is determined only through ablation studies, but why does n=3 lead to performance degradation? The paper merely speculates "due to redundancy" without deeper analysis. Similarly, the choices of sub-image sizes (512×512 for images, 224×224 for videos) and overlap size (16 pixels) lack theoretical or experimental justification. Additional experimental analysis of hyperparameter selection should be provided.

**Questions:**

Refer to the weaknessses.

---

> ### Author Response · Authors · 2025-11-21
> **Response to Reviewer M9mm (Part 1)**
>
> We truly appreciate the reviewers’ positive recognition of our work, including the clear problem motivation, the practical value of our input unification strategy, and the breadth and rigor of our experimental evaluation. Your constructive feedback is highly encouraging and has been invaluable in helping us further strengthen the paper. In response to your comments, we have carefully refined the manuscript as detailed below.
>
> > **Q1: Eliminate ambiguity.**
>
> **A1:** We thank the reviewer for this insightful comment. We agree that framing the difference between images and videos as a "modality gap" is conceptually imprecise, as they both fundamentally belong to the visual modality.
>
> Upon internal review, we have revised the manuscript to reframe this core challenge more accurately as a **"temporal dimension extension problem"** rather than a "modality gap." This change strengthens the theoretical positioning of our work, as the reviewer rightly suggested. All corresponding sections in the paper have been updated to reflect this more precise terminology.
>
> We believe this clarification significantly improves the paper's foundation and thank the reviewer for their constructive feedback.

---

> ### Author Response · Authors · 2025-11-21
> **Response to Reviewer M9mm (Part 2)**
>
> > **Q2.1: Fixed-size sub-images are a common operation in image and video processing.**
>
> **A2.1:** We thank the reviewer for this observation. While we agree that dividing images into fixed-size regions is common, our approach differs fundamentally from both standard Vision Transformer (ViT) patching and traditional independent inference strategies.
>
> **1. Distinction from Standard Methods**
>
> * **vs. Standard ViT Patching:** In a typical ViT, the image is tokenized into very small, fixed-size patches (e.g., $16 \times 16$ pixels) which are flattened and projected. In contrast, our "sub-images" represent much larger distinct regions.
> * **vs. Naive Inference Cropping:** A common engineering solution for high-resolution images is to crop a large input into fixed sub-images (e.g., four $512 \times 512$ crops from a $1024 \times 1024$ image) and process them independently. However, this approach severs the dependencies between crops. As demonstrated in our ablation study, this lack of global modeling significantly harms localization accuracy.
>
> **2. Clarifying the Contribution**
>
> **To be clear**, we do **not** claim that simply partitioning an image into fixed-size sub-images constitutes our key novelty. Instead, our primary contribution lies in the design of the **Relay-Based Attention Framework**, which leverages these partitions to enable more **effective** and **scalable** cross-region information flow.

---

> ### Author Response · Authors · 2025-11-21
> **Response to Reviewer M9mm (Part 3)**
>
> > **Q2.2: Connection with existing methods.**
>
> **A2.2:** We sincerely thank the reviewer for the careful reading and valuable comments. Below we clarify the substantial differences between our Global-Local Relay Attention (GLRA) and existing prompt-tuning methods.
>
> 1. Differences from classical prompt tuning [1,2,4]
>    Standard prompt tuning [1,2,4] inserts a few learnable tokens into a frozen pretrained VLM and relies solely on the original self-attention for interaction. No dedicated global-local relay mechanism exists.
>    In contrast, our GLR tokens + GLRA layer are introduced into a **fully trainable forensic backbone** specifically to enable artifact-preserving, resolution-agnostic global modeling. The explicit relay process across local units and the dedicated relay attention layer are entirely new components absent in [1,2,4].
>
> 2. Detailed distinction from DGL [3] (the most similar prior work)
>    DGL [3] simply concatenates global/local prompts and processes them via ordinary self-attention in a frozen CLIP-like model for retrieval ranking. It lacks any relay mechanism and cannot handle arbitrary resolutions without resizing/distortion.
>
>    Our GLRA instead:
>    - Partitions input into local units
>    - Uses GLR tokens to **explicitly relay** structured global cues across units in a sparse, efficient way
>    - Targets **dense prediction** (pixel/frame-level mask localization) rather than retrieval
>    - Naturally supports **resolution-agnostic** and **unified image/video** processing
>
>    To remove any ambiguity, we conducted **strictly controlled replacement experiments** (same backbone, training recipe, hyperparameters) by replacing GLRA with the exact DGL global-local attention mechanism:
>
>    Table 1. **MVSS Protocol** (F1↑, GFLOPs↓)
>
>    | Method          | Coverage | Columbia | NIST16 | CASIA | IMD2020 | Avg    | GFLOPs       |
>    |-----------------|----------|----------|--------|-------|---------|--------|--------------|
>    | DGL attention   | 0.3077   | 0.4272   | 0.1549 | 0.5566| 0.2751  | 0.3444 | 754          |
>    | Ours (RelayViT) | **0.5510**| **0.7620**| **0.3350**| **0.7400**| **0.3810**| **0.5538**| **119/238/476** |
>
>    DGL variant **severely overfits** and drops ~20 points on average.
>
>    Table 2. **CAT Protocol**
>
>    | Method          | Coverage | Columbia | NIST16 | CASIA | Avg    | GFLOPs |
>    |-----------------|----------|----------|--------|-------|--------|--------|
>    | DGL attention   | 0.6274   | **0.8946**   | 0.4538 | 0.7918| 0.6919 | 754    |
>    | Ours (RelayViT) | **0.6470**| 0.8775   | **0.4756**| **0.8059**| **0.7015**| **119/238/476** |
>
>    Even when performance becomes comparable, DGL requires **1.58× more computation** (+278 GFLOPs absolute). The gap explodes on high-resolution images and long videos.
>
>    Additionally, DGL cannot dynamically adjust computation with resolution while preserving parallelism, whereas GLRA does so seamlessly—an important practical advantage.
>
> In summary, GLRA introduces a **fundamentally new attention paradigm** with three key advantages unavailable in prior prompt-tuning works:
>
> - **Resolution-adaptive computation** (no interpolation/heavy padding) **while preserving parallelism**
> - **Unified spatio-temporal modeling** for images & videos in a single model
> - **Superior efficiency-accuracy trade-off** for high resolution manipulation localization
>
> The new controlled experiments and the above analysis clearly demonstrate that GLRA is **not a minor variant** of DGL or other prompt-tuning methods, but a **novel mechanism** tailored to the unique demands of forensic dense prediction.
>
> We believe these clarifications and additional experiments substantially strengthen the paper’s contribution. Thank you again for the constructive feedback.
>
> **Reference:**
>
> [1] Liu, Xiao, et al. "P-tuning v2: Prompt tuning can be comparable to fine-tuning universally across scales and tasks." arXiv preprint arXiv:2110.07602 (2021).
>
> [2]Zhang, Ji, et al. "Dept: Decoupled prompt tuning." Proceedings of the IEEE/CVF Conference on Computer Vision and Pattern Recognition. 2024.
>
> [3] Yang, Xiangpeng, et al. "Dgl: Dynamic global-local prompt tuning for text-video retrieval." Proceedings of the AAAI Conference on Artificial Intelligence. Vol. 38. No. 7. 2024.
>
> [4] Jia, Menglin, et al. "Visual prompt tuning." European conference on computer vision. Cham: Springer Nature Switzerland, 2022.

---

> ### Author Response · Authors · 2025-11-21
> **Response to Reviewer M9mm (Part 4)**
>
> > **Q3: Improve readability.**
>
> **A3:** We appreciate the feedback on clarity and have revised the manuscript as follows:
>
> 1.  **Punctuation:** We corrected the phrasing in Lines 103–104 and 152–153 to be more direct and formal.
> 2.  **Related Work:** We substantially restructured this section. Rather than listing prior work, we now provide a critical categorization that analyzes limitations in existing approaches. We explicitly connect these gaps to our specific design choices to better highlight our motivation.
>
> > **Q4.1 How the dimensions are allocated in RoPE.**
>
> **A4.1:** Thank you for this important comment. In the revised manuscript, we clarified the RoPE design in **Appendix A.5** and provided the justification and ablation below.
>
> **1. Dimension Allocation & Rationale**
>
> We employ an equal allocation scheme where $d_T = d_{id} = d_H = d_W$, with any remaining dimensions unrotated. This follows validated practices in recent high-dimensional RoPE extensions (e.g., RoFormer [1], Qwen2-VL [2]).
>
> A full theoretical derivation or an extensive sensitivity sweep over all possible splits (e.g., highly asymmetric allocations) is an interesting research direction, but it would **substantially expand the paper’s scope**. **It is more appropriately explored in dedicated work.**
>
> **2. Ablation Study**
>
> To isolate the impact of RoPE, we removed it from the GLRA module. As shown below, removing 4D RoPE leads to training instability and consistently lower performance across all benchmarks.
>
> **Table: Impact of 4D RoPE on performance**
> | RoPE | Coverage | Columbia | NIST  | CASIAv1 | Avg.  |
> | :--- | :--- | :--- | :--- | :--- | :--- |
> | ✗    | 0.542    | 0.682    | 0.301 | 0.707   | 0.517 |
> | **✓**| **0.551**| **0.762**| **0.335**| **0.740**| **0.554**|
>
> **References:**
>
> [1] Su, Jianlin, et al. RoFormer: Enhanced Transformer with Rotary Position Embedding. Neurocomputing 568 (2024): 127063.
>
> [2] Wang, Peng, et al. Qwen2-VL: Enhancing Vision-Language Model’s Perception of the World at Any Resolution. arXiv:2409.12191 (2024).

---

> ### Author Response · Authors · 2025-11-21
> **Response to Reviewer M9mm (Part 5)**
>
> > **Q4.2. On the "strong extrapolation capability" claim and empirical evidence**
>
> **A4.2:** The main manuscript **already includes experiments demonstrating the property of extrapolation**, and we will add more detailed explanations in the revised version. In Protocol-MVSS, our model is **trained only at relatively low resolutions**, with:
>
> - **Training set resolution range:**
>   *min:* 240×160,  *max:* 600×901
> - **Test set resolution:** up to **1024×1024** (Table 1 Pixel-level comparison...) and **4K resolution** (Table 7 Impact of interpolation...), which is already significantly outside the observed training range.
>
> This directly evaluates extrapolation. The results in the main paper (Table 1 Pixel-level comparison... and Table 7 Impact of interpolation.) show that the model maintains stable performance even when the resolution exceeds the training regime.
>
> Moreover, (Table 7 Impact of interpolation...) is conducted at **4K resolution**, which is included in the experimental section Sec. 4.3. Despite being far beyond the training resolutions, our model preserves consistent visual capability, supporting the claimed extrapolation capacity.
>
> Besides, RoPE's extrapolation behavior has been well-studied and validated in prior work. For example:
>
> - Qwen2-VL (Wang et al., 2024)[1] explicitly analyzes RoPE's resolution-scaling behavior in multimodal transformers.
> - JAFAR (Couairon et al., NIPS 2025)[2] further demonstrates that RoPE retains stable extrapolation even at extreme resolutions and provides theoretical insights aligned with our design choice.
>
> These works collectively support the empirical foundation behind our claim.
>
> We appreciate the reviewer's comments, which helped us clarify these points and improve the manuscript.
>
> References:
>
> [1] Wang, Peng, et al. Qwen2-VL: Enhancing Vision-Language Model’s Perception of the World at Any Resolution. arXiv:2409.12191 (2024).
>
> [2] Couairon, Paul, et al. "JAFAR: Jack up Any Feature at Any Resolution." *arXiv preprint arXiv:2506.11136* (2025).

---

> ### Author Response · Authors · 2025-11-21
> **Response to Reviewer M9mm (Part 6)**
>
> > **Q5:Hyperparameter selection lacks rigor.**
>
> **A5:** We appreciate the suggestion to rigorous justify hyperparameters. We have expanded our analysis in the **Appendix (Sec. A.6, A.8)** and clarify our choices below:
>
> **1. GLR Token Quantity ($n=2$):**
>
> Visualizations (Fig.6) in **Appendix A.6** demonstrate that $n=2$ tokens naturally divide labor, focusing on complementary spatial regions. Increasing to $n=3$ leads to attention pattern redundancy (overlap in focus) and performance stagnation; thus, $n=2$ is the most efficient choice.
>
> **2. Sub-image Sizes:**
> * **Video ($256 \times 256$):** Our video data is limited to $512 \times 512$. Using a $512$ sub-image size would collapse the mechanism into full self-attention. $256$ is the maximum size that preserves the intended spatial hierarchy.
> * **Images ($512 \times 512$):** This offers the optimal trade-off between discriminative power and efficiency. We added an ablation (Table 1) showing that $512$ significantly outperforms $256$ for high-res image inputs.
>
> **Table 1. Ablation on sub-image size and overlap for image inputs**
> | Coverage | Columbia | NIST  | CASIA | IMD2020 | Avg.  | Overlap | Sub-image |
> | :--- | :--- | :--- | :--- | :--- | :--- | :--- | :--- |
> | 0.389    | 0.658    | 0.260 | 0.685 | 0.355   | 0.469 | w/o     | 256       |
> | 0.434    | 0.645    | 0.261 | 0.709 | 0.337   | 0.477 | w/      | 256       |
> | **0.551**| **0.762**| **0.335**| **0.740** | **0.381** | **0.554** | **w/** | **512** |
>
> **3. Overlap Size (16 px):**
>
> We selected 16px to match exactly **one ViT patch**, since we use ViT-Base-patch16 as the backbone. This is the minimal overlap required to maintain cross-region feature continuity. Increasing overlap yields diminishing returns while quadratically increasing GFLOPs.

---

### Official Review · Reviewer_3x25 · 2025-10-30

**Soundness:** 3
**Presentation:** 4
**Contribution:** 4
**Rating:** 8
**Confidence:** 4

**Summary:**

The paper tackles visual manipulation localization for both images and videos, addressing resolution diversity and modality gaps. It proposes RelayFormer, which partitions inputs into fixed-size sub-images and introduces Global-Local Relay (GLR) tokens to propagate global context via a Global-Local Relay Attention (GLRA) mechanism. The method includes a parameter-efficient strategy using shared backbone layers with task-specific adapters, a 4D RoPE positional encoding, and a lightweight query-based mask decoder. Experiments on multiple image and video benchmarks show improved F1/IoU performance against prior work, with analyses of FLOPs/parameters and robustness.

**Strengths:**

A new framework for unified image and video forensics.

Achieves top average F1 on image benchmarks, and consistently high IoU/F1 across video inpainting methods on MOSE.

Robustness curves for Gaussian blur, noise, and JPEG compression.

**Weaknesses:**

The proposed approach relies on fusing local and global information, a well-established technique in computer vision whose effectiveness is often assumed.

While results from training on both are presented, there is no clear explanation or investigation into how these modalities mutually influence each other. An intuitive hypothesis is that images can be considered single frames of videos, and training on image forgery could enhance video forgery detection (and vice-versa). To truly demonstrate the benefit of a unified approach, it is suggested to include experiments evaluating model performance when trained exclusively on one modality (image or video) and then tested on both. This would shed light on the synergy between the two data types.

**Questions:**

Please see the weaknesses.

---

> ### Author Response · Authors · 2025-11-21
> **Response to Reviewer 3x25 (Part 1)**
>
> We are greatly encouraged that you found the strengths of **"a new framework", "top average F1 and consistently high IoU/F1 across video inpainting methods", and "robustness"**. We sincerely thank you for your valuable suggestions, which certainly help improve our work. We have accordingly refined our paper as follows:
>
> > **Q1: Rely on fusing local and global information.**
>
> **A1:** Thanks for pointing it out
>
> **Justification for the Local-Global Fusion Approach.**
>
> We acknowledge that fusing local and global information is a well-established paradigm. However, we argue that its application in **manipulation localization** presents unique challenges and requirements distinct from general recognition tasks.
>
> 1. **Task-Specific Necessity:**
>
> As detailed in our manuscript (and illustrated in Fig. 2), manipulation detection requires a delicate balance. It demands fine-grained sensitivity to detect subtle local artifacts (e.g., edge inconsistencies) while simultaneously requiring scene-level reasoning to identify semantic contradictions (e.g., illumination mismatches in splicing or structural redundancy in copy-move). Purely local methods miss these semantic cues, while standard global methods are often computationally expensive for high-resolution forensics.
>
> 2. **Empirical Validation (GLRA vs. Full Attention):**
>
> To demonstrate that our specific implementation (GLRA) is not merely "assuming" effectiveness but provides a tangible and efficient solution, we conducted a quantitative comparison against a **Full Self-Attention** baseline (a proxy for theoretical upper-bound global modeling).
>
>    | Method         | Coverage | Columnbia | NIST   | CASIAv1 | Avg.   |
>    | -------------- | -------- | --------- | ------ | ------- | ------ |
>    | Self-Attention | 0.5690   | 0.9012    | 0.4711 | 0.8158  | 0.6893 |
>    | GLRA           | 0.4977   | 0.8908    | 0.4575 | 0.8442  | 0.6726 |
>
>  As shown in the table, the performance gap is marginal: the Full Self-Attention model achieves an average F1 of 0.6893, while our proposed GLRA achieves 0.6726.
>
> This experiment confirms that our "relay" mechanism successfully captures the necessary global dependencies (scene-level regularities) without the prohibitive cost of dense pixel-level correspondence, validating that the proposed fusion is both sufficient and highly efficient for this domain.

---

> ### Author Response · Authors · 2025-11-21
> **Response to Reviewer 3x25 (Part 2)**
>
> > **Q2: How do image and video mutually influence each other?**
>
> **A2:** We thank the reviewer for the insightful comment. Following this suggestion, we conducted controlled ablation studies where the model is trained on images only, videos only, and both jointly. The corresponding results are reported in **Table 4**, with detailed analysis added to **Sec. 4.2 “Image–Video Interaction in Unified Training.”**
>
> | Num. | Training set | COV. | Col. | NIST16 | CASIAv1 | IMD2020 | Splice | MOSE |
> | ---- | ------------ | ---- | ---- | ------ | ------- | ------- | ------ | ----- |
> | 1 | Img+V-All | 0.569 | 0.755 | 0.282 | **0.753** | 0.357 | **0.472** | 0.684 |
> | 2 | Img+V-Spl | **0.569** | 0.756 | 0.282 | 0.753 | 0.357 | 0.476 | 0.090 |
> | 3 | Img+V-Inp | **0.570** | 0.733 | 0.308 | 0.748 | 0.357 | 0.133 | 0.681 |
> | 4 | V-Spl | 0.051 | 0.147 | 0.163 | 0.143 | 0.219 | 0.264 | 0.119 |
> | 5 | V-Inp | 0.005 | 0.139 | 0.066 | 0.029 | 0.061 | 0.003 | **0.688** |
> | 6 | Img | 0.551 | **0.762** | **0.335** | 0.740 | **0.381** | 0.458 | 0.082 |
>
> Based on these experiments, we summarize the mechanism of **Image–Video Joint Training Interaction** and **Image–Video Transfer** as follows:
>
> 1. **Joint Training Synergy Requires Shared Manipulation Categories, and Its Strength Depends on Data Quality and Diversity.**
>    Effective interaction arises only when they share the *same manipulation types*.
>    - **Image -> Video Transfer (Positive):** High-diversity image datasets with clearer spatial artifacts significantly improve video forgery detection (cf. **Exp. 6 vs. 4**). The model leverages robust spatial cues learned from images and transfers them to video frame analysis.
>    - **Video -> Image Transfer (Minimal):** Current video datasets are more homogeneous and contain repetitive artifacts. Adding such video data brings little benefit to image forgery detection, as the weaker source does not provide additional discriminative information.
>
> 2. **No Significant Image–Video Transfer Without Shared Manipulation Types.**
>    When images and videos contain disjoint manipulation categories (e.g., Image Inpainting vs. Video Splicing), the unified training yields limited interaction or benefit (as shown in **Exp. 3 and 5**).
>
> These findings indicate that **Image–Video Joint Training Interaction** is driven by the presence of **shared manipulation categories** and the **relative strength of data quality** between images and videos. Currently, images act as the stronger data source, serving as a reliable anchor that provides transferable supervision to improve video forgery detection.

---

### Official Review · Reviewer_9ZYn · 2025-11-01

**Soundness:** 3
**Presentation:** 4
**Contribution:** 4
**Rating:** 8
**Confidence:** 4

**Summary:**

This paper proposes RelayFormer, a unified framework for visual manipulation localization across both images and videos. It addresses two major challenges: resolution diversity and the image–video modality gap. The key innovation is the Global-Local Relay Attention (GLRA), where Global-Local Relay (GLR) tokens propagate global context efficiently between local sub-images, enabling scalability without interpolation or dense attention. A lightweight query-based decoder produces precise masks. Extensive experiments show state-of-the-art accuracy, strong robustness, and excellent efficiency on multiple benchmarks.

**Strengths:**

- The insight of bridging image and video manipulation localization holds novelty. It will offer a benchmark case for future multi-modal VML tasks.
- The proposed method’s resolution strategy offers a significant computational efficiency advantage within the current field of manipulation detection.
- The experiments in this paper are comprehensive and well-organized, effectively demonstrating the proposed claims.

**Weaknesses:**

- The proposed method shares some similarities with Visual Prompt Tuning[A], as it introduces additional trainable tokens to transmit information and assist decision-making. It is recommended that the authors discuss this connection in an appropriate section of the paper.
- To the best of my knowledge, RoPE is introduced for the first time in a manipulation localization model. The authors may consider analyzing the advantages of this positional embedding for a pure computer vision task like manipulation detection, ideally supported by additional experiments.
- We are currently in an era that values scaling up as an important perspective. Although the authors claim that the proposed structure bridges image and video manipulation localization, the experiments are conducted separately for the two modalities. It would be interesting to explore whether joint training on both images and videos could lead to further improvements or provide additional insights into scaling behavior.

## Reference
- [A] Jia, Menglin, et al. "Visual prompt tuning." European conference on computer vision. Cham: Springer Nature Switzerland, 2022.

**Questions:**

Will the code be publicly available?

---

> ### Author Response · Authors · 2025-11-21
> **Response to Reviewer 9ZYn (Part 1)**
>
> We are greatly encouraged that you found **"the insight holds novelty", "significant computational efficiency advantage", "comprehensive and well-organized experiments"** of our work. We sincerely thank you for your valuable suggestions, which certainly help improve our work. We have accordingly refined our paper as follows:
>
> > **Q1: Connection with *Visual Prompt* *Tuning*.**
>
> **A1:**  Thanks for your valuable suggestions. Indeed, both Visual Prompt Tuning (VPT) and our method utilize a small number of learnable tokens. However, significant differences exist in their **roles, insertion mechanisms, and optimization objectives**. We have added a dedicated discussion in the revised manuscript (see *Appendix A.2*, Line 768-781). The key distinctions are summarized below:
>
> 1. **Different goals and supervision.**
>    VPT aims to adapt pre-trained models for recognition tasks parameter-efficiently. Its prompt tokens mainly serve as **input-level contextual conditioning**, optimized under classification objectives.
>    In contrast, our **Global-Local Relay (GLR) tokens** are optimized with **dense manipulation-localization supervision** (pixel- or region-level masks). Their purpose is to **aggregate and redistribute cross-tile visual information**, which is essential for fine-grained manipulation localization.
>
> 2. **Different functional roles of learnable tokens.**
>    VPT introduces prompts only at the *input* of the transformer layers, acting as static prefix embeddings.
>    Our GLR tokens operate as a **bidirectional relay**:
>    - **Local -> Global:** collecting features from fixed-size sub-regions, and
>    - **Global -> Local:** propagating aggregated global context back to each region.
>      This *two-stage, cross-region relay process* is fundamentally different from VPT’s single-direction prompting mechanism.
>
> 3. **Different insertion patterns and attention behaviors.**
>    VPT prompts behave as additional input tokens.
>    GLR tokens participate in **multiple layers** of attention and explicitly mediate **local-global interactions**, enabling scalable processing of high-resolution images and long videos.
>
> We appreciate the reviewer’s valuable suggestion. The manuscript now contains a dedicated paragraph in the *Appendix A.2* (Line 768-781), which we believe improves the clarity of our contribution.
>
>
> > **Q2: More experiments and analyses on RoPE**
>
> **A2:** We thank the reviewer for the constructive suggestion. As the reviewer noted, our paper is, to the best of our knowledge, the first to introduce **RoPE** into a manipulation-localization model. RoPE has been extensively shown in prior work [1][2] to provide strong **extrapolation capability**, particularly when dealing with variable-resolution or long-range dependencies. Motivated by these properties, we adopt RoPE as the foundation of our modular design, enabling the model to seamlessly handle **arbitrary spatial resolutions** and to extend naturally to **temporal modeling**.
>
> **1. Newly Added Ablation Study on RoPE**
>
> To further address the reviewer’s concern, we performed an additional ablation experiment by removing RoPE entirely from our architecture. The results show clear degradation in performance and larger instability across datasets:
>
> | Coverage | Columnbia | NIST  | CASIAv1 | Avg.  | RoPE |
> | -------- | --------- | ----- | ------- | ----- | ---- |
> | 0.542    | 0.682     | 0.301 | 0.707   | 0.351 | ×    |
> | **0.551**    | **0.762**     | **0.335** | **0.740**   | **0.381** | √    |
>
> The RoPE-enabled model consistently outperforms the version without RoPE. This confirms that RoPE contributes substantially to the robustness and stability of manipulation localization.
> This ablation has been added to the revised manuscript. See Appendix A.8 (Line 1094-1107).
>
> **2. Resolution Extrapolation Experiments**
>
> In Protocol-MVSS, our model is **trained only at relatively low resolutions**:
>
> - **Training resolution range:**
>   *min:* 240×160   *max:* 600×901
> - **Testing resolution:** up to **1024×1024**, which is well outside the training regime.
>
> As shown in the original manuscript (Table 1 Pixel-level comparison), the model maintains **stable performance** even as the input resolution substantially exceeds what was seen during training, directly validating the extrapolation capability enabled by RoPE.
>
> Furthermore, we include additional evaluations at **4K resolution** in the main experiments (See Table 7 Impact of interpolation). Despite being far beyond the training scale, the model preserves consistent detection behavior, providing further empirical support for RoPE’s strong extrapolation properties in a pure computer-vision task.
>
> References:
>
> [1] Couairon, Paul, et al. "JAFAR: Jack up Any Feature at Any Resolution." *arXiv preprint arXiv:2506.11136* (2025).
>
> [2] Wang, Peng, et al. Qwen2-VL: Enhancing Vision-Language Model’s Perception of the World at Any Resolution. arXiv:2409.12191 (2024).

---

> ### Author Response · Authors · 2025-11-21
> **Response to Reviewer 9ZYn (Part 2)**
>
> > **Q3: The effects of training the image and video data together.**
>
> **A3:** Thanks for pointing it out. As suggested, we have conducted extensive experiments involving joint training (Image+Video) versus independent training. The results are reported in the Table A below.
>
> Table A reports the F1 obtained under six training configurations: image-only (Img), video-inpainting-only (V-Inp), video-splice-only (V-Spl), image + video inpainting (Img+V-Inp), and image + video inpainting + video splice (Img+V-All). Our key observations regarding scaling behavior are as follows:
>
> Table A   F1 results of six training settings on various datasets.
> | Num. | Training set | COV.         | Col.         | NIST16       | CASIAv1      | IMD2020      | Splice       | MOSE         |
> | ---- | ------------ | ------------ | ------------ | ------------ | ------------ | ------------ | ------------ | ------------ |
> | 1    | Img+V-All    | 0.569 | 0.755 | 0.282        | **0.753**    | 0.357 | **0.472**    | 0.684        |
> | 2    | Img+V-Spl    | **0.569**    | 0.756        | 0.282 | 0.753 | 0.357        | 0.476        | 0.09  |
> | 3    | Img+V-Inp    | **0.570**    | 0.733        | 0.308 | 0.748 | 0.357        | 0.133        | 0.681 |
> | 4    | V-Spl        | 0.051        | 0.147        | 0.163        | 0.143        | 0.219        | 0.264 | 0.119        |
> | 5    | V-Inp        | 0.005        | 0.139        | 0.066        | 0.029        | 0.061        | 0.003        | **0.688**    |
> | 6    | Img          | 0.551        | **0.762**    | **0.335**    | 0.740        | **0.381**    | 0.458        | 0.082        |
>
> 1.  **Image Data Scales Video Performance:** We observed a strong positive scaling effect when adding image data to video training. As shown in our new experiments (comparing Experiment **6** vs. **4** and **2**), training on images alone significantly outperforms training on videos alone for shared manipulation types (e.g., splicing). Furthermore, adding image data to video training boosts performance, acting as a "Generalization Anchor." This confirms that scaling up training data by including high-quality image forgeries significantly benefits video forgery detection.
>
> 2.  **Asymmetric Benefit:** Interestingly, the reverse scaling effect is not currently observed. Adding video data to image training (Experiments **1, 2, 3**) did not consistently improve image forgery detection. We attribute this to the current quality of video forgery datasets, which lack the diversity and label precision of image datasets.
>
> We have added a new subsection, "Sec 4.2 Interaction Between Image and Video in Unified Training", and a corresponding Table in the revised manuscript to report these findings.
>
> > **Q4: Will the code be publicly available?**
>
> **A4:** Yes,  of course! The source code and the pre-trained models will be released.

---

> > ### Comment · Reviewer_9ZYn · 2025-11-27
> > **Good and inspiring paper.**
> >
> > Thanks to the author's detailed rebuttal, which solved my concerns. The experiment about the interaction between is inspiring for future research. Further enrich the presentation of the paper and match the claim of `unified framework` in the title. Overall, I keep my rating of 8: good paper for ICLR.

---

### Author Response · Authors · 2025-11-25
**Rebuttal for feedback**

We would like to thank the reviewer for the helpful discussion during the first round of the review. We hope our response has adequately addressed your concerns. We view this as a great opportunity to improve our work and shall be grateful for any additional feedback you could provide.

---

### Author Response · Authors · 2025-12-02
**Post-Rebuttal Summary for AC**

Dear Area Chair,

We are deeply grateful for the time and energy you have dedicated to this process, particularly given the unique challenges and heavy workload of the current ICLR cycle.

To assist with your decision-making, here's a brief overview of each reviewer's post-rebuttal status:

1. **Reviewer 9ZYn (Rating: 8 / Confidence: 4)** Initially requested clearer links to Visual Prompt Tuning, analysis of RoPE, and stronger support for the unified framework. After the rebuttal **effectively addressed all concerns**, the reviewer praised the work as **inspiring for future research** and maintained a positive score.

2. **Reviewer 3x25 (Rating: 8 / Confidence: 4)** highlighted the need to analyze cross-modality interactions and validate the unified framework through cross-modality experiments, which is similar to Reviewer 9ZYn, and we conducted **extensive additional experiments** to address these concerns.

3. **Reviewer M9mm (Rating: 4 / Confidence: 4)**: Despite this reviewer's **lack of engagement during the rebuttal**, we have fully addressed the reviewer’s concerns regarding novelty by demonstrating that GLRA achieves **better performance** than the DGL baseline with **only 63% of the computational cost**. Regarding the requested experiments on extrapolation and hyperparameters, we clarified that some results were **already present in the original manuscript**, while the remaining requested data has now been **comprehensively added to the appendix**. We also revised the manuscript according to the reviewers' requirements.

4. **Reviewer k3yb (Rating: 6 -> 8 / Confidence: 5)** primarily questioned the potential information bottleneck of relay tokens and suggested extending the evaluation to the CAT protocol. We resolved these concerns through additional experiments that validated our design choices and robustness, leading the reviewer to **raise their score following our response, prior to the unfortunate information leak incident**.

   (**Because Reviewer k3yb directly revised the score without leaving any comments, the final score is unfortunately not reflected in the system due to the change in ICLR rebuttal policy.**)

Thank you again for your time. We hope the above summary helps streamline your decision process.

Best regards,

The Authors

---

### Meta-Review · Area_Chair_ZVBB · 2026-01-07

**Summary:**

The reviews have broad consensus that the paper is well motivated, technically sound, and empirically thorough. The authors rebuttal and follow up discussions are detailed. Main discussion points center on the degree of methodological novelty relative to prior prompt-based or global local designs, the conceptual framing of image video unification, and whether the claimed trade-offs and benefits are empirically substantiated. The rebuttal effectively addressed most substantive concerns. Remaining concerns are outweighed by the demonstrated empirical gains, efficiency advantages, and reviewer confirmations that key questions were satisfactorily answered.

My recommendation is Accept (poster), justified by 2 strong accept scores with acknowledged rebuttal resolution, 1 marginally positive but well-supported review, and 1 weaker novelty-focused review whose core technical concerns were largely mitigated during discussion.

**Reviewer Concerns:**

The rebuttal discussions are quite detailed and in depth. Questions about similarity to prompt-tuning and global local baselines were clarified. Requests for stronger empirical support of the unified image video framework were addressed with new cross-training studies and CAT-protocol results. Concerns about relay token information loss and RoPE design were also addressed through added ablations.

**Reviewer Scores:**

Reviewer 9ZYn: explicitly confirmed that the rebuttal resolved their concerns and stated he/she would keep the score; no change expected.

Reviewer 3x25: Given that the requested cross-modality experiments were added and directly addressed the main weakness, the score would likely remain at 8.

Reviewer M9mm: Given other reviews with high scores, and new ablations, likely he/she will increase to a slightly higher score.

Reviewer k3yb: The rebuttal directly addressed all technical questions with new experiments, and the reviewer indicated an increased score.

---

### Decision · Program_Chairs · 2026-01-26

Accept (Poster)